# G-KV: Decoding-Time KV Cache Eviction with Global Attention

## Abstract

Recent reasoning large language models (LLMs) excel in complex tasks but encounter significant computational and memory challenges due to long sequence lengths. KV cache compression has emerged as an effective approach to greatly enhance the efficiency of reasoning. However, existing methods often focus on prompt compression or token eviction with local attention score, overlooking the long-term importance of tokens. We propose G-KV, a KV cache eviction method that employs a global scoring mechanism, combining local and historical attention scores to more accurately assess token importance. Additionally, we introduce post-training techniques, including reinforcement learning and distillation, to optimize models for compressed KV cache settings. The code of this paper is available on: `https://anonymous.4open.science/r/G-KV-B3C0`.

## 1 Introduction

Large language models (LLMs) have garnered widespread attention and applications. Recently released reasoning models have demonstrated remarkable performance (Guo et al., 2025; Team et al., 2025; Yang et al., 2025), even in addressing complex tasks such as mathematics and coding. These reasoning models achieve significant improvements across various problems through long chain-of-thought (CoT) (Wei et al., 2022), enabling iterative reflection and verification. However, the long CoT of reasoning models typically consists of thousands or even tens of thousands of tokens. This imposes a substantial increase in computational costs and KV cache memory consumption. Notably, the computation of attention becomes a critical bottleneck, as its complexity scales quadratically with the sequence length.

To overcome the bottlenecks of memory and computational complexity, numerous optimization methods for KV cache or attention mechanisms have been proposed (Li et al., 2024a). Among these, some methods prune the KV cache of tokens, significantly reducing computational overhead and memory consumption. However, most of these methods concentrate on the compression of the prompt's KV cache at the prefilling stage (Li et al., 2024b; Cai et al., 2024; Feng et al., 2024; Kim et al., 2025). For reasoning tasks, where the output length often far exceeds the input length, limiting compression efforts to the prompt's KV cache results in only marginal benefits.

Although some methods support dynamically evicting tokens during the decoding stage (Song et al., 2025; Cai et al., 2025), thereby maintaining consistently low KV cache requirements throughout the generation process, they rely solely on the attention scores of a few most recently generated tokens to determine which tokens to evict. However, our experiments reveal that the importance of tokens can shift during the generation process. Such a localized perspective overlooks the long-term significance of tokens. In addition, the original models may fail to adapt to the sparse attention patterns induced by KV cache compression, resulting in suboptimal performance. Xiao et al. (2023) and Chen et al. (2025) train models with sparse attention through pre-training. However, the cost of pre-training is exceedingly high.

To address the limitations of the local attention, **(1)** we propose a **simple and effective global score**. This global score combines the local attention score with historical scores to assess the long-term importance of tokens, thereby avoiding the eviction of critical context that may reappear in future attention patterns. The global score can be seamlessly integrated with most existing methods and significantly enhances performance. Furthermore, **(2)** to enable the original model to adapt to the sparse attention pattern, we implement a **reinforcement learning framework specifically tailored**

**for KV cache compression**, which eliminates the discrepancy between the training policy and the inference policy. **(3)** Our experiments show that integrating the global score under a 512-token budget improves other methods by 5% to 20%. The RL framework we developed for KV cache compression is better suited for training models with compressed KV cache, achieving significantly superior performance compared to RL conducted directly on Full KV cache.

## 2 RELATED WORK

KV cache compression methods can be broadly categorized into four classes (Li et al., 2024a): KV cache selection, merging (Kim et al., 2023; Nawrot et al., 2024; Liu et al., 2024a), quantization (Yao et al., 2022; Sheng et al., 2023; Liu et al., 2024b), and low-rank decomposition (Chang et al., 2024; Dong et al., 2024). KV cache selection is the most pertinent to our work.

**Prefilling KV Cache Compression.** SnapKV (Li et al., 2024b) and KVzip (Kim et al., 2025) determine which KV cache to retain by leveraging the attention score from an observation window or an appended specially designed prompts. PyramidInfer (Yang et al., 2024) and PyramidKV (Cai et al., 2024) allocate varying KV cache budgets across different layers. Ada-KV (Feng et al., 2024) and HeadKV (Fu et al., 2024) propose allocating different budgets to individual attention heads. These methods predominantly focus on compressing the prompt. However, as the reasoning length continues to increase, merely compressing the prompt still faces computational and memory bottlenecks.

**Decoding-time KV cache Eviction.** H2O (Zhang et al., 2023) uses the accumulated attention received by each token as its score, but this can easily lead to the tail tokens being ignored in long sequences. Song et al. (2025) dynamically evicts tokens during decoding using the local attention score of the latest tokens, while R-KV (Cai et al., 2025) introduces redundancy scores to further enhance the information density of the KV cache. Nevertheless, these methods are constrained to attention within local windows. Although CAKE (Qin et al., 2025) considers temporal attention shifts, it remains restricted to a local window.

## 3 PRELIMINARY

Dynamic token eviction methods (Cai et al., 2025; Song et al., 2025) can be conceptualized within a unified framework. In this framework, the KV cache is compressed after every $s$ tokens are generated. The most recent $w$ generated tokens constitute the *observation window*. The query states $\mathbf{Q} \in \mathbb{R}^{h_{\mathrm{q}} \times w \times d}$ of tokens in the observation window, alongside the cached key states $\mathbf{K} \in \mathbb{R}^{h_{\mathrm{kv}} \times l \times d}$ are employed to compute score using a function $f(\mathbf{Q}, \mathbf{K}) : \mathbb{R}^{h_{\mathrm{q}} \times w \times d} \times \mathbb{R}^{h_{\mathrm{kv}} \times l \times d} \to \mathbb{R}^{h_{\mathrm{kv}} \times l}$. Here, $h_{\mathrm{q}}$ and $h_{\mathrm{kv}}$ denote the number of heads for the query states and key states, respectively, while $l$ represents the length of the KV cache. For each head, the key states and value states corresponding to the top-$(b - w)$ scores are retained. Here, $b$ denotes the budget size of the KV cache. Incorporating the KV cache from the observation window of length $w$, the final compressed KV cache has a total length of $b$. **This framework effectively balances memory efficiency with the preservation of critical contextual information for future token generation.**

Typically, these methods involve computing the attention scores between the query states within the observation window and the cached key states. The $i$-th head attention score formula is as follows:

$$\mathbf{A}_{[i,:,:]} = \mathrm{softmax}\left(\frac{\mathbf{Q}_{[i,:,:]}\mathbf{K}_{[j,:,:]}^{\top}}{\sqrt{d}}\right), \tag{1}$$

where $\mathbf{A} \in \mathbb{R}^{h_{\mathrm{q}} \times w \times l}$. Here, $j$ represents the head index for the key states. For multi-head attention (Vaswani et al., 2017), $j = i$. However, in the case of multi-query (Shazeer, 2019) or group-query (Ainslie et al., 2023) attention, $j$ and $i$ exhibit a one-to-many relationship. To obtain the scores corresponding to each key state, a max-reduce operation is performed across the scores of multiple attention heads corresponding to each key head, resulting in $\mathbf{A}' \in \mathbb{R}^{h_{\mathrm{kv}} \times w \times l}$. The final scores $\mathbf{S}$ are then computed from $\mathbf{A}'$ by applying a mean operation within the observation window,

$$\mathbf{S}_{i,j} = \frac{1}{w}\sum_{k=0}^{w-1}\mathbf{A}'_{i,k,j}, \tag{2}$$

yielding $\mathbf{S} \in \mathbb{R}^{h_{kv} \times l}$. This score reflects the significance of the key states and value states within the KV cache.

## 4 OBSERVATION

Dynamic token eviction methods are based on an intuitive assumption: tokens attended by the observation window are the most critical for subsequent generation. The strong performance of these methods suggests that this assumption holds some validity. However, **can a single observation window effectively determine which tokens are truly essential for subsequent generation?** To investigate this question, we devise the following experiment.

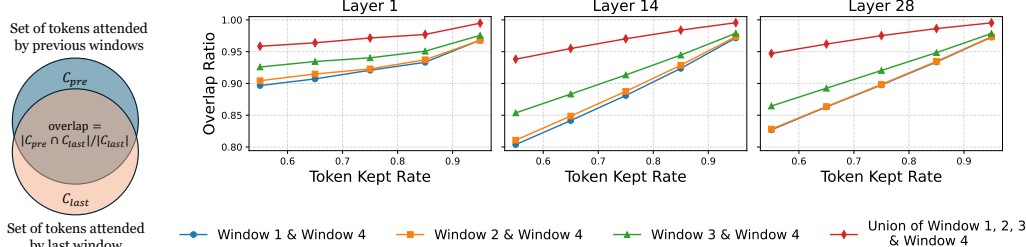

Figure 1: The left figure illustrates the calculation process for overlap. The right figure depicts the overlap between the last window and other windows across different layers. The horizontal axis represents the proportion of tokens retained.

We conducted inference using DeepSeek-Distill-Qwen-7B (Guo et al., 2025) on the AMC 2023 benchmark (AoPS, 2023), performing 32 rollouts per question. The last 512 tokens of the generated output were divided into 4 observation windows, and for each window, the score $\mathbf{S}$ (equation (2)) was calculated based on the query status of tokens within the window and the key status of preceding tokens. Based on $\mathbf{S}$, the $p\%$ tokens with the highest scores were identified as the most attended tokens for each window. Subsequently, we quantified the *overlap* between the token set attended to by the last observation window and those attended to by other windows, defined as the ratio of the intersection size to the size of the token set attended to by the last window. The results, shown in Figure 1, reveal that:

**Observation 1** *The tokens attended to by the last window are not fully consistent with those attended to by the earlier windows. As the number of retained tokens decreases, the inconsistency becomes more pronounced.*

This finding demonstrates that the importance of tokens shifts across different windows. **If KV cache compression is performed based on scores computed from a single window, some tokens that possess long-term importance are likely to be prematurely evicted due to being temporarily overlooked by a single window.**

Furthermore, we computed the overlap between the last window and the union of all preceding windows (the red line in Figure 1). We find that:

**Observation 2** *The overlap between the token set attended to by the last window and the union of tokens attended to by all preceding windows is relatively higher. Notably, even when retaining only 55% of the tokens, the overlap approaches 95%.*

This observation further illustrates that the attention received by tokens is **intermittent**. On the other hand, it also indicates that **tokens receiving significant attention are highly likely to have been similarly attended to by at least one preceding observation window.**

## 5 TRAINING-FREE KV CACHE COMPRESSION WITH GLOBAL ATTENTION

As previously discussed, scores computed from a single window are insufficient to effectively capture the long-term importance of tokens. To address this limitation, we aim to determine which tokens should be evicted by leveraging their attention scores across a broader context.

The characteristics of human memory reveal that memories revisited multiple times become increasingly reinforced, whereas those left unreviewed for extended periods gradually diminish. Inspired by this, we propose a global score to quantify the degree to which tokens are attended to throughout the decoding process. We introduce a **memory decay rate**, $\alpha \in [0, 1]$, to promote the eviction of tokens that no longer attract attention. We experiment with three different forms for calculating the global score: *max*, *average*, and *summation*. For $i < b - w$, the three different forms of global scores are calculated using the following formulas:

$$\mathbf{F}_t[:, i] = \max \left( \alpha \cdot \mathbf{F}_{t-1}[:, i], \frac{\mathbf{S}_t[:, i]}{\max_j(\mathbf{S}_t[:, j])} \right), \tag{3}$$

$$\mathbf{F}_t[:, i] = \alpha \cdot \mathbf{F}_{t-1}[:, i] + (1 - \alpha) \cdot \frac{\mathbf{S}_t[:, i]}{\max_j(\mathbf{S}_t[:, j])}, \tag{4}$$

$$\mathbf{F}_t[:, i] = \alpha \cdot \mathbf{F}_{t-1}[:, i] + \frac{\mathbf{S}_t[:, i]}{\max_j(\mathbf{S}_t[:, j])}, \tag{5}$$

Here, $\mathbf{F}_{t-1} \in \mathbb{R}^{h_{kv} \times (b-w)}$ represents the historical global scores from the previous step, while $\mathbf{F}_t \in \mathbb{R}^{h_{kv} \times l}$ denotes the global scores in the current step. The attention scores $\mathbf{S}_t$ (equation (2)) are normalized by the maximum values within each attention head. Since only $b - w$ tokens have scores from the previous step, for $i \geq b - w$, $\mathbf{F}_t = \frac{\mathbf{S}_t[:, i]}{\max_j(\mathbf{S}_t[:, j])}$. Based on $\mathbf{F}_t$, we select $b - w$ tokens whose corresponding KV cache is retained, and the $\mathbf{F}_t$ values of the retained tokens are subsequently recorded for use in the next compression step. At the first compression step, as $\mathbf{F}_{t-1}$ is not available, KV cache selection is performed directly based on $\mathbf{S}_t$.

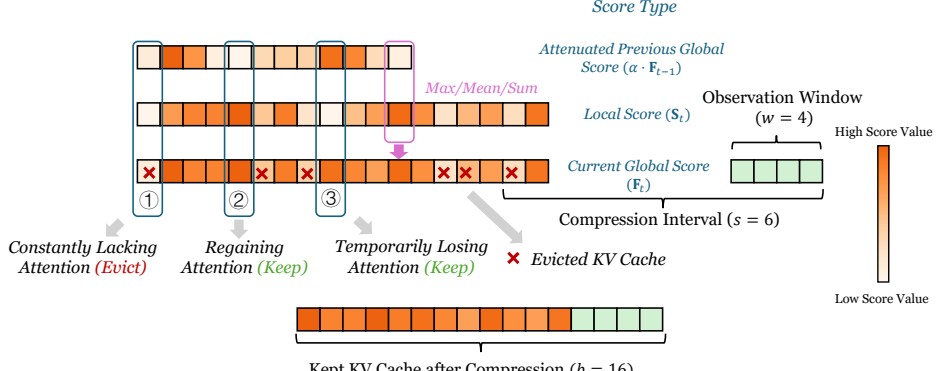

Figure 2: This figure illustrates the computation process of the global score. Each block represents the KV cache of a token, with the block's color indicating its score (darker color represents higher scores).

The score $\mathbf{F}_t$ takes into account both the attention received in the current observation window and the attention from preceding windows, and we refer to it as the **global score**. In contrast, $\mathbf{S}_t$, computed within a single window, is referred to as the **local score**. Figure 2 illustrates several scenarios that emerge when utilizing the global score:

- **Low $\mathbf{F}_{t-1}[i, j]$ and low $\mathbf{S}_t[i, j]$:** This implies that the $i$-th KV head's $j$-th token consistently receives very little attention across multiple consecutive windows. Such tokens are considered insignificant and are therefore eligible for eviction.

- **Low $\mathbf{F}_{t-1}[i,j]$ and high $\mathbf{S}_t[i,j]$:** This suggests that the token temporarily lost attention in the previous observation window but regained attention in the current window. These tokens are re-engaged in the ongoing context.

- **High $\mathbf{F}_{t-1}[i,j]$ and low $\mathbf{S}_t[i,j]$:** This signifies that the token, while not receiving attention in the current window, was highly attended to in previous windows. Unlike other methods that might immediately evict such tokens, we choose to retain them because these tokens are highly likely to be attended to again in the future.

Compared to the local score, the global score better reflects the long-term importance of a token. In addition, our analysis reveals that even with the KV cache compression algorithm, the attention scores remain highly sparse, as detailed in Appendix B. Each observation window focuses on only a small subset of tokens within the compressed KV cache. **By employing global scores, the compression algorithm retains a small subset of tokens that are highly attended to by each observation window. Furthermore, tokens that are likely to receive significant attention in future windows are highly likely to be included within the union of these subsets.**

## 6 Enhancing KV Cache Compression through Training

Dynamic token eviction can be seen as a form of sparse attention, where the KV cache of evicted tokens becomes inaccessible to subsequent tokens. Figure 3 shows the sparse attention mask corresponding to the KV cache compression process. We define the original policy (LLMs) as $\pi_\theta$, with $\theta$ representing model parameters, and the policy with KV cache compression or sparse attention as $\pi'_\theta$. The original model $\pi_\theta$ is trained with full attention, relying on complete context. After compression, the policy $\pi'_\theta$ operates in a constrained context environment with unchanged parameters $\theta$, making $\pi'_\theta$ sub-optimal in this setting.

We aim to enable the model to adapt to the condition of KV cache compression. We explore post-training methods for this purpose. Specifically, we implemented a reinforcement learning (RL) framework that **supports generation with KV cache compression and training with sparse attention masks**. In this framework, sampling is performed directly by the policy $\pi'_\theta$. During generation, the positions of the tokens actually evicted are recorded and used to construct the sparse attention mask. The optimization objective is as follows:

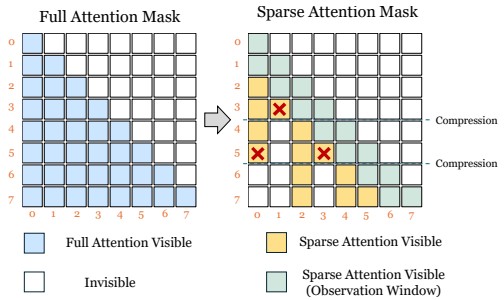

Figure 3: Illustration of the sparse attention mask. If the block at the $i$-th row and $j$-th column is visible, it indicates that the $j$-th token in the sequence can attend to the $i$-th token. Red crosses represent tokens evicted during a KV cache compression process; these tokens are invisible to newly generated tokens in subsequent steps.

$$\mathcal{J}(\theta) = \mathbb{E}_{\{y_i\}_{i=1}^G \sim \pi'_{\theta_{\text{old}}}(\cdot|x)} \left[ \frac{1}{G} \sum_{i=1}^G \frac{1}{|y_i|} \sum_{t=1}^{|y_i|} \min\left( r_{i,t}(\theta)\hat{A}_{i,t}, \text{clip}(r_{i,t}(\theta), 1-\epsilon, 1+\epsilon)\hat{A}_{i,t} \right) \right], \quad (6)$$

where $r_{i,t}(\theta) = \frac{\pi'_\theta(y_{i,t}|x,y_{i,<t})}{\pi'_{\theta_{\text{old}}}(y_{i,t}|x,y_{i,<t})}$ and $\hat{A}_{i,t} = \frac{r_i - \text{mean}(\{r_j\}_{j=1}^G)}{\text{std}(\{r_j\}_{j=1}^G)}$. Here, $r_i$ represents the reward associated with the response $y_i$. For each input $x$, we perform $G$ times sampling, where $y_{i,t}$ denotes the $t$-th token in the output of the $i$-th sampling. This is the optimization objective of GRPO (Shao et al., 2024) without KL regularization. Moreover, training on outputs truncated due to the maximum output length constraint may introduce interference (Yu et al., 2025). To address this, we directly set their advantages to zero. Nevertheless, this RL method may only be suitable for tasks where the rewards of outputs can be easily verified. Consequently, we propose a more general distillation-like method, as detailed in Appendix C.

## 7 EXPERIMENT

### 7.1 EXPERIMENT SETUP

**Benchmark and Dataset**. We evaluate the model's reasoning capabilities in the domains of mathematics and coding. For the mathematics domain, we employ AMC 2023 (AoPS, 2023) and AIME 2024 (AoPS, 2024) as benchmarks. AMC is designed for middle-school students as an entry-level mathematical competition, while AIME serves as a critical gateway to advanced mathematics contests, featuring more challenging problems. For the coding domain, we conduct evaluations on LiveCodeBench (Jain et al., 2024), which includes programming competition problems of varying difficulty levels. For RL training, we use the DeepScaleR-40k (Luo et al., 2025) dataset, which incorporates mathematical problems of varying difficulty levels from different datasets. Additionally, 27k correct reasoning-based responses are sampled from the DeepScaleR-40k dataset using DeepSeek-R1-Distill-Qwen-7B, and these samples are utilized for distillation.

**Model**. We evaluate our approach using DeepSeek-R1-Distill-Qwen-7B and DeepSeek-R1-Distill-Llama-8B (Guo et al., 2025). These models are reasoning models distilled from DeepSeek-R1 using Qwen 2.5 (Team, 2024) and LLaMA 3.1 (Grattafiori et al., 2024), respectively.

**Evaluation Protocol and Metrics**. For sampling, we set the temperature to 0.6 and the top-p parameter to 0.95. Unless otherwise specified, for the AMC 2023, the maximum sequence length is configured to 16k, while for the AIME 2024, it is set to 32k. We use pass@1 (Chen et al., 2021) as our evaluation metric, which is an unbiased estimate of the probability that the model answers a question correctly on the first attempt. For each question, we perform sampling 32 times to estimate the pass@1 score.

### 7.2 THE ABLATION AND COMPARISON OF GLOBAL SCORE

In this section, we conduct experiments on different forms of global scores and various values of $\alpha$, comparing them with the Local Score. Additionally, **CAKE** (Qin et al., 2025) uses the attention variance within a local window to represent the degree of attention fluctuation, which we refer to as the attention shift score, and we compare our method against it.

**Implementation Details**. We set the KV cache budget to 512 ($b = 512$), the observation window size to 16 ($w = 16$), and perform compression after generating every 128 tokens ($s = 128$).

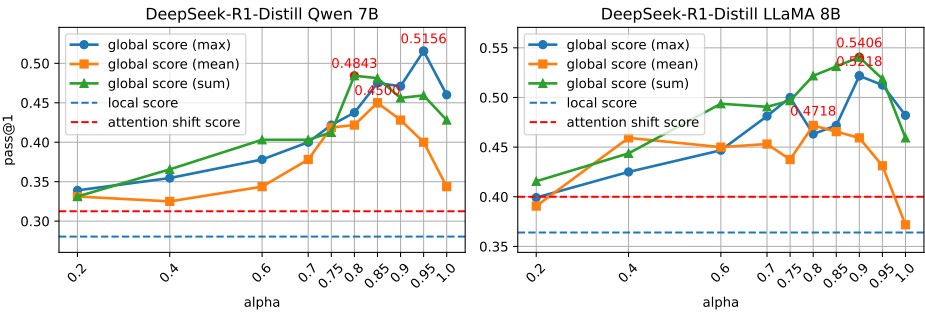

Figure 4: Performance of different methods on the AMC 23 benchmark.

**Analysis**. As shown in Figure 4, all three forms of global score deliver significant improvements and perform notably better than the attention shift score in CAKE. However, the mean form of the global score performs slightly worse than the other two. The performance of all three methods remains relatively stable when $\alpha \in [0.8, 0.9]$, achieving notable performance gains. This range is recommended as the optimal hyperparameter setting.

### 7.3 MAIN RESULTS

In this section, we evaluate the performance of various methods under different budget constraints. We refer to the method that combines the global score (max) with the redundancy score (Cai et al.,

2025) as **G-KV**, and we set $\alpha$ to 0.8. Appendix D describes how to combine global score with other methods. The baselines for comparison include StreamingLLM (Xiao et al., 2023), SnapKV (Li et al., 2024b), Local Score, and R-KV (Cai et al., 2025). For StreamingLLM, the budget refers to its window size. For SnapKV and R-KV, the parameters $s$ and $w$ are consistent with those used in the previous section, while other parameters follow the settings specified in their respective papers. Additionally, we report the average **Token Retention Ratio**, defined as the ratio of the KV Cache length to the total sequence length. This ratio is calculated exclusively for cases where the model generates the **correct** answer. **A lower token retention ratio indicates the model can function properly with longer generation lengths under a fixed budget.**

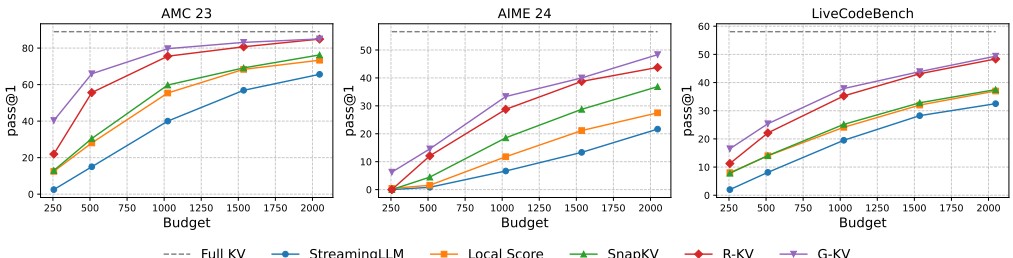

Figure 5: Performance of different compression methods with DeepSeek-R1-Distill Qwen 7B model.

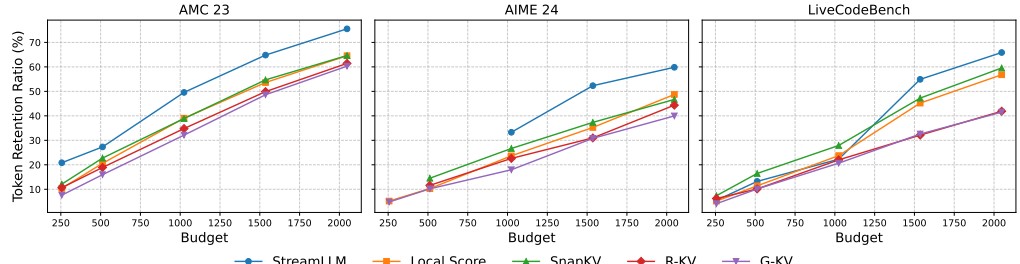

Figure 6: Token retention ratio of different compression methods with DeepSeek-R1-Distill Qwen 7B model.

**Analysis**. As shown in Figure 5, our method achieves SOTA performance across most budgets and benchmarks. The fewer the budget tokens, the greater the advantage of our method over others. For the AMC 23 benchmark, our approach achieves nearly a 20% improvement under a 512-token budget. The results in Figure 6 also demonstrate that our method achieves the lowest token retention ratio in most scenarios. This indicates that the tokens retained by our method have a higher information density, enabling the model to function effectively on longer sequences.

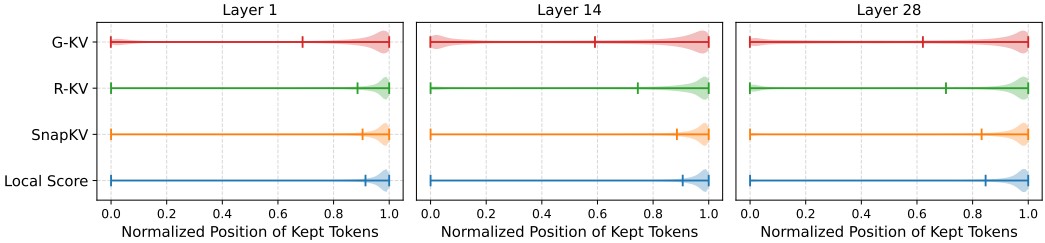

Figure 7: The density distribution of the normalized final retained token positions for different algorithms using DeepSeek-R1-Distill-Qwen-7B. The results are evaluated on the AMC 23 benchmark. The vertical bars in the figure indicate the **mean values**.

Furthermore, we normalize the positions of tokens retained by different algorithms (with budget of 512) within the complete sequence and visualize their density distributions, as shown in Figure 7.

The tokens retained by previous methods based on local scores are concentrated towards the end of the sequence. This phenomenon arises because the context near the observation window during each compression step tends to have higher semantic similarity, compounded by the inherent characteristics of RoPE (Su et al., 2024). In contrast, when using global scores, the retained token positions are more evenly distributed, allowing more comprehensive information to be preserved. This characteristic may explain why G-KV performs significantly better than other methods in handling longer generation sequences and under lower budget constraints. A cases presented in Appendix J illustrate this phenomenon more intuitively. We explain this phenomenon as arising from the fact that tokens close to the observation window are more likely to receive higher attention scores. Consequently, methods based on local scores tend to prioritize retaining tokens nearest to the observation window. In contrast, attention to more distant contexts is often intermittent. Global scores allow these intermittently attended but important tokens to be retained, whereas local scores are more prone to evict them when their attention scores temporarily drop.

## 7.4 RESULTS OF TRAINING

In this section, we further present the results obtained using different training methods. We refer to our proposed RL and distillation methods as **RL-Sparse** and **Distill**, respectively. For comparison, we include reinforcement learning conducted with generation and training with the Full KV cache, which we refer to as **RL-Full**.

**Implementation Details.** The training is based on the DeepSeek-R1-Distill-Qwen-7B model. For RL training, the maximum output length is 4096, with a sampling temperature of 0.6. Each step samples 16 questions, with 8 responses generated per question, yielding 128 trajectories for gradient computation and updates in a single batch (allowing gradient accumulation via micro-batches). RL training runs for 400 steps, rewarding 1 for correct responses and 0 for incorrect ones. For distillation, the maximum sequence length is 4096, with longer sequences truncated. Training is performed for 250 steps (around 1 epoch) with a batch size of 128. The learning rate is set to $1 \times 10^{-6}$. The G-KV method is used for RL-Sparse with a budget of 2048, while other parameters remain as previously mentioned. All evaluations in this section are restricted to an output length of 4096, consistent with the training setup.

| | AMC 23 | | | AIME 24 | | |
|---|---|---|---|---|---|---|
| **Budget** | **512** | **1024** | **2048** | **512** | **1024** | **2048** |
| **Untrained** | 45.00 | 54.21 | 59.84 | 11.56 | 18.64 | 23.02 |
| **Distill** | 47.89 (+2.89) | 56.48 (+2.27) | 61.48 (+1.64) | **14.27** (+2.71) | 21.56 (+2.92) | 24.79 (+1.77) |
| **RL-Full** | 47.65 (+2.65) | 56.79 (+2.58) | 63.82 (+3.98) | 12.18 (+0.62) | 21.14 (+2.50) | 25.93 (+2.91) |
| **RL-Sparse** | **51.01** (+6.01) | **61.71** (+7.50) | **67.65** (+7.81) | 13.75 (+2.19) | **22.18** (+3.54) | **26.66** (+3.64) |

Table 1: Pass@1 comparison across different training methods and budgets on AMC 23 and AIME 24.

**Analysis**. We evaluated the trained models under different budgets, with the results summarized in Table 1. **RL-Sparse** achieves the best performance across most settings, significantly outperforming models trained with the Full KV cache. By directly optimizing the policy $\pi'_\theta$, **RL-Sparse** minimizes the training-inference discrepancy, resulting in superior performance. In contrast, **RL-Full** shows moderate gains but is hindered by the mismatch between its training policy $\pi_\theta$ and the inference policy. The distillation method effectively enables $\pi'_\theta$ under constrained KV cache to approximate $\pi_\theta$, offering a practical alternative for scenarios where verifiable reward functions are difficult to design. Additional training information and analysis are provided in Appendix E.

## 7.5 EFFICIENCY ANALYSIS

In this section, we analyze the efficiency of the KV cache compression algorithm. Throughput is used as the evaluation metric, calculated as the total number of valid tokens generated (excluding padding tokens) divided by the time consumed. We extracted 1,024 mathematical problems from the DeepScaleR-40k dataset and conducted inference with varying batch sizes, using a maximum output length of 16k. All experiments were performed on a single A100 GPU.

| Batch Size | DeepSeek-Qwen-Distill-7B | | | DeepSeek-Llama-Distill-8B | | |
|---|---|---|---|---|---|---|
| | **32** | **64** | **128** | **16** | **32** | **64** |
| **Full-KV** | 62.41 | OOM | OOM | 31.08 | OOM | OOM |
| **R-KV (Budget 2048)** | 172.44 | 203.66 | 238.43 | 82.33 | 98.01 | 111.89 |
| **G-KV (Budget 512)** | 261.32 | 475.59 | 760.74 | 158.43 | 517.30 | 612.60 |
| **G-KV (Budget 1024)** | 212.93 | 367.96 | 448.35 | 118.10 | 193.56 | 258.18 |
| **G-KV (Budget 2048)** | 170.64 | 221.23 | 248.22 | 93.91 | 118.82 | 154.52 |

Table 2: Throughput comparison (tokens/s). OOM refers to the occurrence of an Out of Memory error, indicating insufficient GPU memory.

**Analysis.** As shown in Table 2, our method achieves a significant improvement in throughput compared to Full-KV under the same batch size. For DeepSeek-Qwen-Distill-7B, throughput improves by $4.18\times$, $3.41\times$, and $2.73\times$ under KV cache budgets of 512, 1024, and 2048, respectively. Similarly, DeepSeek-Qwen-Llama-8B achieves throughput gains of $5.09\times$, $3.79\times$, and $3.02\times$ under the same budgets. Naturally, the reduced memory requirements of the KV cache allow inference with larger batch sizes. For these two models, the throughput of GKV reaches up to $12.18\times$ and $19.7\times$ that of Full-KV, respectively.

Additionally, we conducted experiments on the throughput of R-KV. Since our method operates within the same framework as R-KV and introduces minimal additional computation, the difference in throughput between the two methods is negligible.

In addition to throughput, decoding time is a critical factor influencing user experience in practical applications. As shown in Figure 8, the decoding time under different KV cache compression budgets is similar and, at the same batch size, is approximately 40% of that of Full-KV. Further comparisons and analyses of decoding time are provided in Appendix F. Additionally, we conducted an analysis of memory

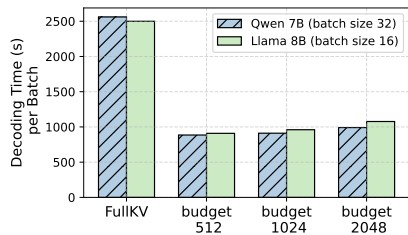

Figure 8: Decoding time comparison.

efficiency. Under the 16k context setting, our method achieves approximately a 90% reduction in KV cache memory usage. Detailed results are provided in Appendix G.

| Budget | 512 | 1024 | 2048 |
|---|---|---|---|
| Local score | 46.1 | 74.5 | 117.6 |
| Global score | 50.5 | 79.4 | 121.4 |

Table 3: Average Compression Time (ms)

| Budget | 512 | 1024 | 2048 |
|---|---|---|---|
| Local score | 0.77% | 1.10% | 1.59% |
| Global score | 0.83% | 1.20% | 1.57% |

Table 4: Compression Time Ratio (%)

We use DeepSeek-Distill Qwen-7B to measure the average compression time per step for global score and local score under a batch size of 32. The experimental results are shown in Table 3. Global score introduces an additional delay of approximately 5 ms per compression step. However, this delay is negligible in the context of the entire decoding process. Table 4 presents the proportion of compression time relative to the total decoding time, showing that the compression process for both global score and local score accounts for only about 1% of the total time.

# 8 CONCLUSION

In this paper, we propose G-KV, a KV cache compression method that integrates local and historical attention scores to assess token importance globally. In addition, post-training techniques, including reinforcement learning and distillation, are introduced to adapt LLMs to compressed KV cache settings. Experiments on AMC-23 and AIME-24 benchmarks confirm effectiveness of G-KV. G-KV significantly reduces memory and computational bottlenecks, enabling efficient and scalable reasoning for LLMs.

## REPRODUCIBILITY STATEMENT

The code associated with this paper is available at: `https://anonymous.4open.science/r/G-KV-B3C0`. It includes the necessary environment configurations and execution scripts. All datasets and benchmarks utilized in this study are publicly accessible. The distilled 27k data used in our experiments is provided as part of the supplementary materials.

## ETHICS STATEMENT

This work does not involve any human participants, personally identifiable information, or sensitive data. No experiments were conducted on animals or humans. Therefore, we declare that there are no ethical concerns associated with this study.

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

## A    USE OF LLMs

We utilized ChatGPT-4o [1] to refine the content based on our original writing. All revised text was subsequently reviewed and verified by us. The architecture of the code was designed by our team, with Claude-4 [2] assisting in the implementation of certain functional components. All code has undergone comprehensive testing to ensure its reliability.

## B    THE SPARSE NATURE OF ATTENTION

In this section, we analyze the sparsity of attention scores. Specifically, let the maximum attention score in a sequence be $s_{max}$. We define $p \times s_{max}$, where $p \in (0, 1)$, as a threshold. Tokens with attention scores below this threshold are considered to receive minimal attention. The *sparsity* is defined as the proportion of tokens with attention scores below the threshold relative to the total number of tokens in the sequence.

For the full KV cache, we compute the attention scores between the query states of the last 16 tokens and the key states of all preceding tokens. However, we only evaluate the sparsity of the last 512 tokens. For KV cache compression algorithms, we calculate the attention scores between the query states of tokens in the last observation window and the key states retained in the kept KV cache. For all KV cache compression algorithms, we set the budget to 512, the observation window size to 16, and the compression interval to 128.

Figure 9 illustrates the sparsity levels across different models and layers. For full KV cache, the attention scores of most layers exhibit extremely high sparsity. In the majority of layers, over 90% of tokens have attention scores lower than 1% of the maximum score. This observation indicates that most tokens are not attended to by the last few tokens, which also serves as the primary motivation behind the design of most existing KV cache compression algorithms (Zhang et al., 2023; Cai et al., 2025).

Furthermore, we conducted an analysis of sparsity when applying KV cache compression algorithms. Although the sparsity decreases significantly compared to the full KV cache, notable sparsity still persists. For DeepSeek-R1-Distill-Qwen-7B, many layers still exhibit over 80% of tokens having attention scores below 5% of the maximum score. Similarly, for DeepSeek-R1-Distill-LLaMa-8B, with the exception of the first layer, more than 90% of tokens in other layers have attention scores below 1% of the maximum score. **This indicates that even after KV cache compression, the attention scores between the compressed KV cache and the observation window still maintain a high degree of sparsity. This means that each observation window still only attends to a subset of tokens within the compressed KV cache.**

---

[1]https://chatgpt.com
[2]https://claude.ai

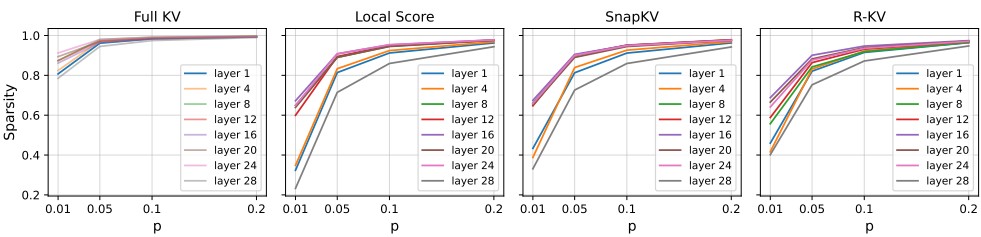

(a) Sparsity of attention in DeepSeek-R1-Distill-Qwen-7B.

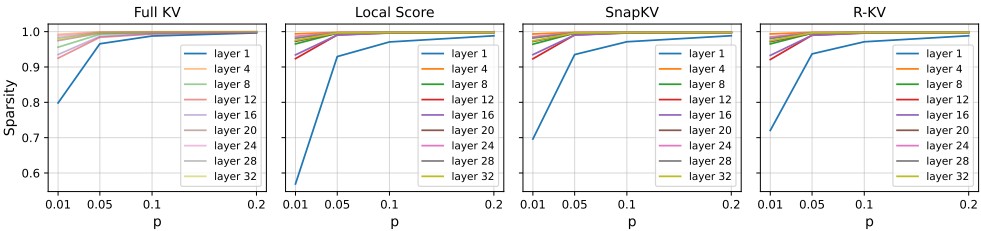

(b) Sparsity of attention in DeepSeek-R1-Distill-LLaMA-8B.

Figure 9: Comparison of attention sparsity. The horizontal axis represents the coefficient $p$ multiplied by the maximum attention score to calculate the threshold. The vertical axis represents the sparsity.

## C   DISTILLATION-LIKE TRAINING WITH SPARSE ATTENTION MASK

As discussed previously, when defining the reward for outputs becomes challenging, reinforcement learning (RL) may no longer be applicable. In such scenarios, alternative training methods need to be explored. In this section, we propose a distillation-based approach.

Specifically, we sample outputs $y \sim \pi_\theta(y|x)$ from $\pi_\theta$. Then, we simulate the execution of our KV cache eviction algorithm to determine which tokens would be evicted during the generation process, thereby constructing a corresponding sparse attention mask. We train $\pi'_\phi$ ($\phi$ initialized as $\theta$) with the sparse attention mask through a soft target loss (Hinton et al., 2015):

$$\mathcal{L}(\phi) = \tau^2 \, \mathrm{KL}\big(\pi_\theta(y|x) \,\big\|\, \pi'_\phi(y|x)\big), \tag{7}$$

where $\tau$ represents the sampling temperature, and $\mathrm{KL}(\cdot\|\cdot)$ denotes the Kullback-Leibler (KL) divergence (Kullback & Leibler, 1951). Minimizing this objective allows the distribution of the policy employing KV cache compression to approximate that of the full KV cache policy. This training approach also enables the model to adapt to the sparse attention.

## D   INTEGRATING THE GLOBAL SCORE WITH OTHER METHODS

The global score is an inherently versatile technique that can be seamlessly integrated into other methods. We take SnapKV (Li et al., 2024b) and R-KV (Cai et al., 2025) as examples to demonstrate the results after incorporating the global score. The schematic diagrams of these methods are illustrated in Figure 10.

SnapKV introduces sequence-wise max-pooling helps to retain more detailed information from the prompt. When the global score is integrated with SnapKV, it suffices to replace the local score utilized by SnapKV with the global score. Figure 11 illustrates the effect of combining SnapKV with the global score (max). Replacing the local score with the global score effectively improves the performance of SnapKV; however, it does not surpass the performance of using the global score alone. This may be attributed to the pooling mechanism of SnapKV, which is designed for prefilling-stage

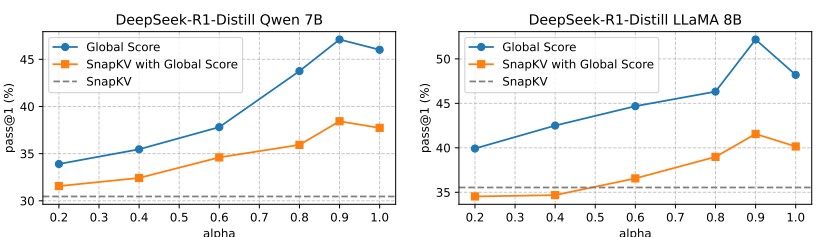

Figure 10: This figure illustrates the computation process of local score, SnapKV and R-KV. Each block represents the KV cache of a token, with the block's color indicating its score (darker color represent higher scores).

Figure 11: The performance on AMC 23 of SnapKV with Global Score.

compression to retain more detailed information from the prompt. However, during the decoding phase, this design may hinder the eviction of less important tokens.

R-KV proposed a redundancy score to identify redundant tokens in the KV cache. By removing these redundant tokens, it becomes possible to retain more informative content within a limited KV cache budget. Specifically, the cosine similarity between the Key states, $\mathbf{K} \in \mathbb{R}^{h_{\mathrm{kv}} \times l \times d}$, is calculated as follows:

$$\overline{\mathbf{K}}_{i,j} = \frac{\mathbf{K}_{i,j}}{\|\mathbf{K}_{i,j}\|_2 + \epsilon},$$

$$\mathbf{C}_i = \overline{\mathbf{K}}_i(\overline{\mathbf{K}}_i^T).$$

Here, $\mathbf{C} \in \mathbb{R}^{h_{\mathrm{kv}} \times l \times l}$, $\mathbf{C}_i$ represents the cosine similarity between the key states of the $i$-th attention head. Cai et al. (2025) mask the elements in $\mathbf{C}$ below a specific threshold to zeros, as well as those corresponding to the most recent tokens, resulting in a modified similarity matrix $\mathbf{C}'$. The average similarity score for each token is computed as:

$$\overline{\mathbf{C}}'_{i,j} = \sum_{k=0}^{l-1} C'_{i,k,j}.$$

Here, $\mathbf{C}'_{i,j}$ represents the redundancy level of the $j$-th token in the $i$-th attention head. A higher value of $\mathbf{C}'_{i,j}$ indicates that the token is more redundant. Finally, the redundancy score $\mathbf{R}$ is obtained by applying the softmax function to the average similarity scores:

$$\mathbf{R}_{i,j} = \frac{\exp(\overline{\mathbf{C}}'_{i,j})}{\sum_{k=0}^{l-1} \exp(\overline{\mathbf{C}}'_{i,k})}.$$

In Equation (5), we perform max normalization on the local scores. This approach is adopted because, as the sequence length increases, the attention distribution becomes diluted, and the average magnitude of attention scores for each token changes. Previous methods did not account for the combination of scores across windows, and thus normalization was unnecessary. In contrast, we mitigate

this issue by applying max normalization to the local scores. The redundancy scores calculated via the softmax function also suffer from a similar dilution problem. Therefore, when combining our global scores with the redundancy scores, we also normalize the redundancy scores as follows:

$$\mathbf{R}'_{i,j} = \frac{\mathbf{R}_{i,j}}{\max_j \mathbf{R}_{i,j}}$$

where $\mathbf{R}'_{i,j}$ represents the normalized redundancy score. Finally, we combine the global score and the redundancy score using the following formula:

$$\mathbf{F}'_t = \lambda \cdot \mathbf{F}_t - (1 - \lambda) \cdot \mathbf{R}'$$

where $\lambda \in [0, 1]$ is a weighting factor that determines the relative contribution of the global score and the redundancy score.

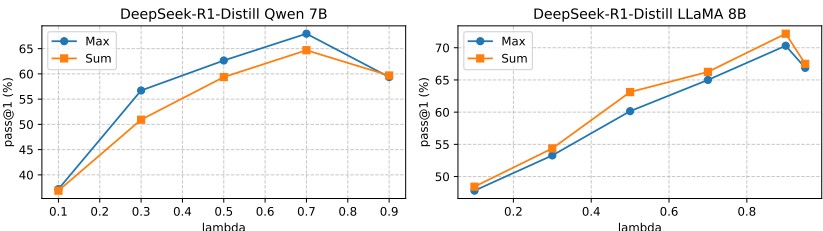

Figure 12: Results of combining global scores with redundancy scores under different $\lambda$ values for DeepSeek-R1-Distill Qwen-7B (left) and Llama-8B (right).

Since we normalize the redundancy scores, we re-tune the hyperparameter $\lambda$ instead of directly adopting the value $\lambda = 0.1$ as used in the original paper. We fix $\alpha$ at 0.8. Figure 12 presents the experimental results of combining global scores (max and sum) with redundancy scores under different $\lambda$ values. For DeepSeek-R1-Distill Qwen-7B, the best performance is achieved when $\lambda = 0.7$, while for DeepSeek-R1-Distill Llama-8B, the optimal performance is observed at $\lambda = 0.9$. In both cases, the global score plays a dominant role.

## E    MORE INFORMATION AND ANALYSIS OF TRAINING

In this section, we visualize certain information recorded during the training process. Figure 13 illustrates the average KL divergence between the distributions of the sparse model $\pi'_\theta$ and the full attention model $\pi_\theta$ for generating the next token during distillation training. As training progresses, the KL divergence decreases rapidly, indicating that the distribution of $\pi'_\theta$ is indeed approaching that of $\pi_\theta$.

Figure 14 (a) and (b) depict the changes in entropy (Shannon, 1948) and pass@1 on the validation set during reinforcement learning training. The validation set consists of 32 samples randomly selected from the training set. For each question in the validation set, pass@1 is estimated by sampling 4 times.

Entropy reflects the uncertainty of a distribution. The curves in Figure 14 (a) indicate that RL-Sparse exhibits relatively higher entropy during the reinforcement learning process, which is due to the fact that the sparse attention mask introduces some information loss. However, as training progresses, the entropy of both RL-Sparse and RL-Full decreases rapidly. This suggests that the determinism of generation distribution from policy trained by RL-Sparse and RL-Full is improving. However, overly high determinism might lead to insufficient exploration. For larger-scale reinforcement learning, it would be beneficial to integrate advanced techniques to encourage more exploration (Liao et al., 2025; Cui et al., 2025).

The curve of pass@1 in Figure 14 (b) demonstrates that RL-Sparse achieves even higher pass@1 compared to RL-Full. It is worth noting that RL-Sparse is evaluated under a compressed KV cache

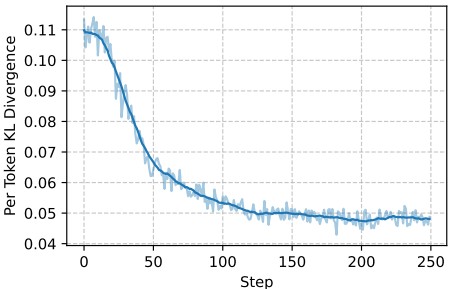

Figure 13: The average per-token KL divergence between the sparse model $\pi'\theta$ and the full attention model $\pi\theta$ during distillation training. The semi-transparent curves represent the actual values, while the solid lines indicate the smoothed values.

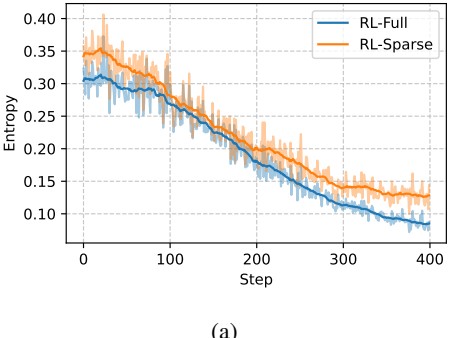

(a)

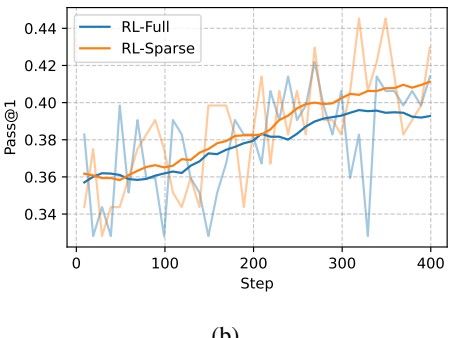

(b)

Figure 14: Changes in entropy (left) and pass@1 on the validation set (right) during reinforcement learning training. The semi-transparent curves represent the actual values, while the solid lines indicate the smoothed values.

setting, while RL-Full is evaluated with a full KV cache. **The superior performance of RL-Sparse in terms of pass@1 may indicate the potential for faster convergence in sparse reinforcement learning.** As highlighted by Wang et al. (2025), backpropagating gradients selectively on high-entropy tokens during reinforcement learning training yields better results. Sparse RL might **focus gradient updates more effectively on tokens with higher information density and greater decision-making significance, thereby improving the efficiency of policy optimization.** We believe this offers new insights for future research on reinforcement learning for LLMs.

# F  BALANCE BETWEEN THROUGHPUT AND DECODING TIME

In §7.5, we compared the decoding time under the same batch size. The experimental results at that time indicated that the differences in decoding time across various budgets were minimal. This was due to the insufficient batch size, which failed to fully utilize the computational units of the GPU. In this section, we further analyze and compare the decoding times across different budgets under varying batch sizes.

As illustrated in Figure 15, an increase in batch size leads to longer decoding times. Moreover, with higher budgets, the differences in decoding time across varying batch sizes become more pronounced. For instance, when the budget is set to 512, the decoding time for a batch size of 128 is only marginally greater than that for a batch size of 32. However, when the budget increases to 2048, the decoding time surpasses more than twice the initial value. Therefore, while KV cache compression supports larger batch sizes and larger batch sizes typically yield higher throughput. However, To avoid excessively long decoding times, batch sizes should not be set excessively large.

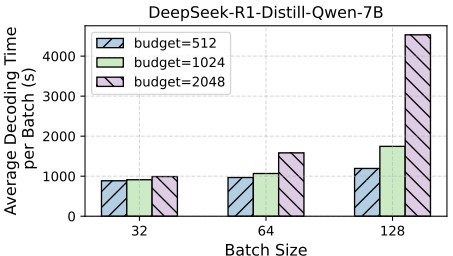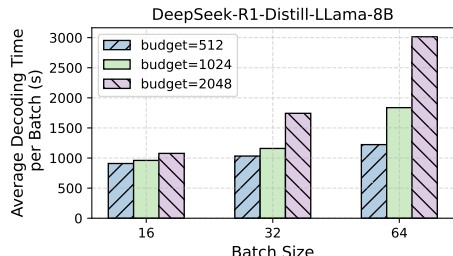

Figure 15: Comparison of decoding times across different budget and batch size.

## G  MEMORY ANALYSIS

**KV Cache Memory Analysis.** We take the DeepSeek-R1-Distill-Qwen-7B model as an example, which has 28 layers, an attention head dimension of 128, and 2 key-value heads. Assuming a sequence length of 16,384 (16k) and use precision of bf16 (2 bytes), the memory consumption of key and value per sequence can be calculated as $(28 \times 128 \times 2 \times 16384 \times 2 \times 2)/2^{30} \approx 0.44$ GB. When the batch size is 128, the KV cache alone requires 56 GB of memory. However, when applying KV cache compression, the required KV cache memory is reduced to $\frac{b+s}{\text{sequence length}}$, where $b$ denotes the KV cache budget and $s$ is the interval between two consecutive compression operations. For a batch size of 128, $s = 128$ and budgets of 512, 1024, and 2048, the KV cache memory requirements are reduced to 2.18 GB, 3.93 GB, and 7.43 GB, respectively. This results in memory savings of 96.1%, 92.9%, and 86.7%, respectively.

**Score Cache Memory Analysis.** Our method stores the scores computed for compression, denoted as $\mathbf{F} \in \mathbb{R}^{h_{\text{kv}} \times (b-w)}$, while the shape of the key or values status cache is $\mathbf{K}, \mathbf{V} \in \mathbb{R}^{h_{\text{kv}} \times b \times d}$. The memory consumption of the scores, as a fraction of the KV cache memory, is $\frac{b-w}{b \times d \times 2}$, where $w \ll b$. This simplifies to approximately $\frac{1}{2 \times d}$. Since $d$ is typically 128, the additional overhead from storing the scores is negligible compared to the memory savings achieved. Under the same settings as in the example above, with a fixed KV cache budget of 2048, the KV cache size is approximately 7 GB, while the global score cache occupies around 27 MB.

**Sparse Attention Mask Memory Analysis.** The memory consumption of sparse attention masks is easily overlooked; however, in practice, it can surpass even the size of the model parameters. Consider a scenario where the training batch size per device (GPU) is 16, the model consists of 28 layers, $h_{\text{kv}} = 2$, the sequence length is 4096 (4k), and the data type of sparse attention mask occupies only 1 byte. The memory required for the sparse attention mask is calculated as $(16 \times 28 \times 2 \times 4096 \times 4096)/2^{30} = 14$ GB. Loading the complete sparse attention mask onto the GPU may lead to out-of-memory (OOM) errors. To address this, the sparse attention mask is offloaded to the CPU after its construction. Furthermore, during training, **we employ gradient checkpointing and ensure that only the sparse mask for a single layer is loaded onto the GPU at any given time**. This strategy is critical for enabling training with sparse attention masks.

## H  EXPERIMENTAL RESULTS ON DEEPSEEK-R1-DISTILL-LLAMA-8B

Although §4 and §7.3 only present the overlap analysis and the normalized positional density map of retained tokens for the DeepSeek-R1-Distill-Qwen-7B model, similar phenomena are observed for the DeepSeek-R1-Distill-LLaMa-8B model. The corresponding experimental results are shown in Figure 16 and Figure 17.

Figures 18 and 19 present the experimental results of DeepSeek-R1-Distill Llama-8B. When the budget is sufficient, both R-KV and G-KV achieve results comparable to or even surpassing those of Full KV. Under lower budgets, G-KV exhibits a certain advantage.

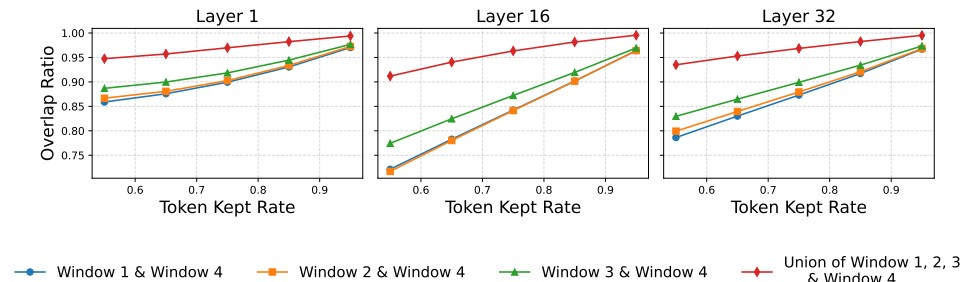

Figure 16: This figure illustrates the overlap between the set of tokens attended to in the last window and the sets of tokens attended to in other windows. The horizontal axis represents the proportion of tokens retained.

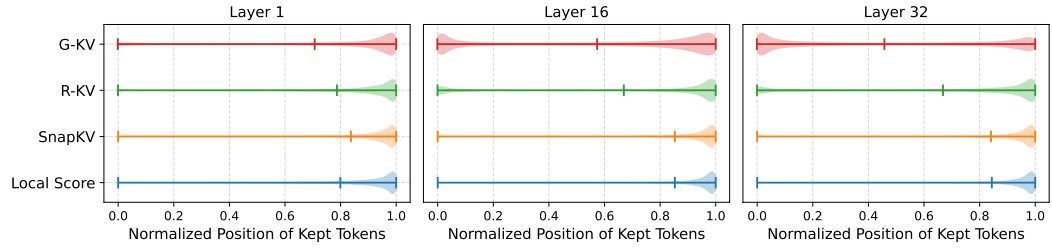

Figure 17: The density distribution of the normalized final retained token positions for different algorithms using DeepSeek-R1-Distill-LLaMa-8B. The results are evaluated on the AMC 23 benchmark. The vertical bars in the figure indicate the **mean values**.

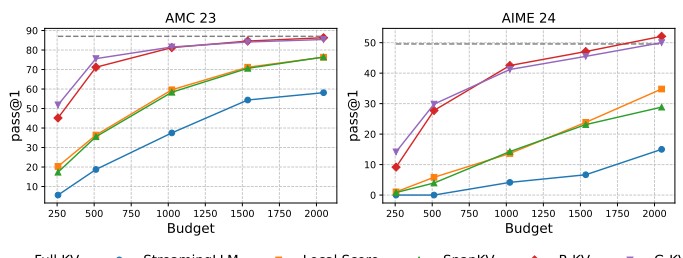

Figure 18: Performance of different compression methods with DeepSeek-R1-Distill LLaMA 8B model.

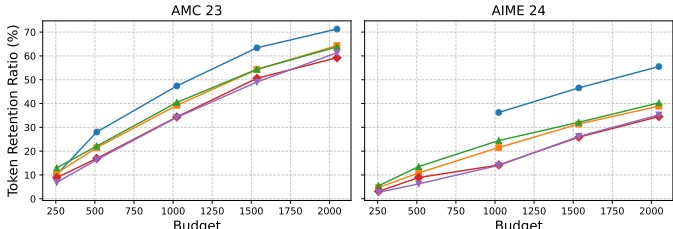

Figure 19: Token retention ratio of different compression methods with DeepSeek-R1-Distill LLaMA 8B model.

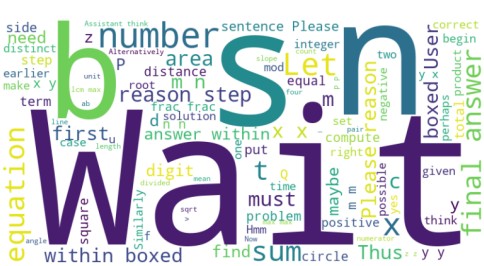 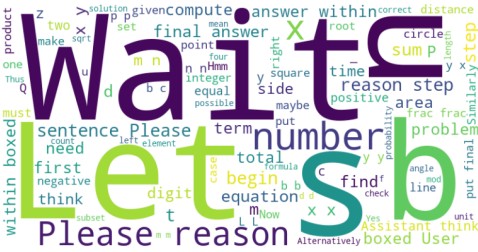

(a) DeepSeek-R1-Distill-Qwen-7B     (b) DeepSeek-R1-Distill-LLaMa-8B

Figure 20: Word Clouds of the kept token on AMC 23

# I DISCUSSION AND INSIGHTS

In this section, we first visualize the tokens retained by a single KV head in the final layer using word clouds. The visualization results are presented in Figure 20. Notably, the word "wait" appears most prominently in both models. This word typically appears when the model begins to engage in reflection. Shah et al. (2025) and Muennighoff et al. (2025) have found that inserting "wait" into the output can significantly enhance the triggering of explicit reflection and improve the final accuracy.

Interestingly, combining the cases presented in Appendix J, we observe that the model's attention is not primarily focused on the reflective content following the word "wait," but instead is remarkably concentrated on the word "wait" itself. This phenomenon suggests that the key and value states corresponding to "wait" may have already encoded the forthcoming reflective information in advance. **In other words, the actual "thinking" process is likely to occur during the compression of information when the model accesses the value states of "wait" through the attention mechanism.**

This implies that "wait" in the deep hidden representations of LLMs carries a sufficiently high semantic density, and the subsequent reflective output merely serves to externalize the content that has already been "pre-thought" within "wait." Of course, the processes of information compression and decoding are not confined only to "wait" but rather constitute a dynamic process throughout the decoding stage. Other tokens (e.g., periods, contrastive conjunctions, etc.) exhibit similar functionalities in deeper layers (Chen et al., 2025): through their representations, these tokens trigger the model to extract, organize, or reconstruct key information.

This further explains why deep-layer attention often exhibits high sparsity: in these layers, the KV cache representations of certain critical tokens already serve as highly compressed semantic carriers. By selectively attending to these tokens, the model can effectively accomplish the contextual integration needed for inference, without exhaustively referencing every prior token. **Interestingly, this behavior mirrors fundamental characteristics of human cognition**. When processing complex information, humans typically do not distribute their attention uniformly across all available details. Instead, they selectively focus on a small number of salient cues, which act as anchors or triggers for downstream reasoning and memory retrieval. For example, in recalling a past experience, one may only need to remember a single vivid scene—such as a spoken phrase or a specific gesture—to reconstruct the broader narrative context.

This mechanism of sparse activation and efficient recall is not merely a cognitive shortcut but a defining feature of the human memory system. It underscores a key insight: *compression as intelligence*. This principle highlights the profound role of semantic compression in enabling efficient reasoning and memory retrieval. **Furthermore, it strengthens the argument for designing and training models with explicit sparsity mechanisms. Such mechanisms not only enhance computational efficiency but may also promote the development of more human-like capacities for abstraction and generalization.**

## J    Case Study

In this section, we provide a case utilizing the global score ($\alpha = 0.8$) and another case employing the local score. The KV cache budget is set to 512. The questions used are sourced from AMC 23, and the model employed is DeepSeek-R1-Distill-Qwen-7B. Tokens retained by a single attention head in the final layer are highlighted in orange, while discarded tokens are indicated in black. Tokens that remain uncompressed after the last compression step are marked in green.

These two cases more intuitively illustrate the insights reflected in Figure 7 and Figure 17. When utilizing the local score, the retained tokens are concentrated towards the last part of the sequence, with most of the earlier tokens being discarded. In contrast, when employing the global score, the retained tokens are more evenly distributed throughout the entire sequence. Interestingly, a significant portion of the prompt is also preserved under the global score, which may explain its work properly on longer sequences.

## Local Score Case

Please reason step by step, and put your final answer within \boxed{}.<|User|> Cities $A$ and $B$ are $45$ miles apart. Alicia lives in $A$ and Beth lives in $B$. Alicia bikes towards $B$ at 18 miles per hour. Leaving at the same time, Beth bikes toward $A$ at 12 miles per hour. How many miles from City $A$ will they be when they meet?<|Assistant|><think>

Okay, so I have this problem where Alicia and Beth live in two cities, A and B, which are 45 miles apart. Alicia starts biking from city A towards city B at 18 miles per hour, and Beth starts biking from city B towards city A at 12 miles per hour. They leave at the same time, and I need to figure out how many miles from city A they will meet. Hmm, okay, let me try to break this down step by step.

First, I know that when two objects are moving towards each other, their relative speed is the sum of their individual speeds. So, Alicia is going at 18 mph and Beth at 12 mph, so together, they're closing the distance between them at a rate of 18 + 12 = 30 mph. That makes sense because they're moving towards each other, so their speeds add up.

Now, the total distance between the two cities is 45 miles. So, if they're moving towards each other at a combined speed of 30 mph, I can calculate the time it takes for them to meet by dividing the distance by their combined speed. Let me write that down:

Time = Distance / Speed = 45 miles / 30 mph.

Calculating that, 45 divided by 30 is 1.5 hours. So, they'll meet after 1.5 hours. Okay, that seems straightforward.

But wait, the question asks how many miles from city A they will meet. So, I need to figure out how far Alicia has traveled in that 1.5 hours because she started from city A. Since she's biking at 18 mph, I can multiply her speed by the time to find the distance.

Distance from A = Speed * Time = 18 mph * 1.5 hours.

Let me compute that. 18 times 1 is 18, and 18 times 0.5 is 9. So, 18 + 9 is 27. So, Alicia has traveled 27 miles from city A when they meet.

Alternatively, I can check this by calculating how far Beth has traveled in the same 1.5 hours and then adding their distances to make sure it totals 45 miles. Beth is biking at 12 mph, so:

Distance from B = 12 mph * 1.5 hours = 18 miles.

So, Alicia has gone 27 miles from A, and Beth has gone 18 miles from B. 27 + 18 is 45 miles, which matches the total distance between the cities. That checks out, so it must be correct.

Wait, another way to think about it is to set up equations based on their positions as functions of time. Let me try that approach to verify.

Let's denote the time until they meet as t hours. The distance Alicia covers is 18t, and the distance Beth covers is 12t. Since they're moving towards each other, the sum of these distances should equal the total distance between the cities, which is 45 miles.

So, 18t + 12t = 45.

Combining the terms on the left, 30t = 45.

Solving for t, t = 45 / 30 = 1.5 hours, which is the same result as before.

Therefore, Alicia has traveled 18 * 1.5 = 27 miles from city A when they meet. So, that's consistent with the first method.

I can also visualize this on a distance-time graph. The x-axis would be time in hours, and the y-axis would be distance from city A. Alicia's line would have a slope of 18 mph, starting at (0,0), and Beth's line would have a slope of −12 mph (since she's moving towards A), starting at (0,45). The point where the two lines intersect would be the time and location where they meet.

Setting up the equations:

For Alicia: y = 18x

For Beth: y = 45 − 12x

Set them equal to each other to find the intersection point:

18x = 45 − 12x

Adding 12x to both sides: 30x = 45

Dividing both sides by 30: x = 45 / 30 = 1.5 hours.

Then, plugging back into Alicia's equation: y = 18 * 1.5 = 27 miles.

Same answer again. So, whether I use relative speed, distance equations, or a graphical approach, the result is 27 miles from city A.

Figure 21: Part 1 of the case of local score.

I wonder if there's another way to think about this problem. Maybe using ratios? Since they're moving towards each other, the ratio of their speeds is 18:12, which simplifies to 3:2. That means Alicia covers 3 parts of the distance, and Beth covers 2 parts. The total parts are 3 + 2 = 5 parts. So, each part is 45 / 5 = 9 miles. Therefore, Alicia covers 3 * 9 = 27 miles, and Beth covers 2 * 9 = 18 miles. That also gives the same answer.

So, using the ratio method, I still get 27 miles from city A. It seems consistent across different methods.

Is there a way to use algebra to model this without time? Hmm, maybe, but I think the methods I've used so far are the most straightforward.

Alternatively, if I let the distance from A be x, then the distance from B would be 45 - x. Since they start at the same time and meet at the same time, the time taken by Alicia to travel x miles is equal to the time taken by Beth to travel (45 - x) miles.

Time is equal to distance divided by speed, so:

Time for Alicia: x / 18

Time for Beth: (45 - x) / 12

Since these times are equal:

x / 18 = (45 - x) / 12

Cross-multiplying:

12x = 18(45 - x)

12x = 810 - 18x

Adding 18x to both sides:

30x = 810

Dividing both sides by 30:

x = 810 / 30 = 27

So, that's another method which gives x = 27 miles. Yep, that's consistent too.

So, regardless of the approach|relative speed, distance equations, ratios, or algebraic time equality|they all lead me to the same answer: 27 miles from city A.

I think I've covered all the methods I can think of. Each time, the result is 27 miles. Therefore, I can confidently say that when Alicia and Beth meet, they will be 27 miles away from city A.

**Final Answer**
They will meet \boxed{27} miles from City A.
</think>

To determine how many miles from City A Alicia and Beth will meet, we can use the concept of relative speed and distance equations.

1. **Relative Speed Method:**
   - Alicia's speed: 18 mph
   - Beth's speed: 12 mph
   - Combined speed: \(18 + 12 = 30\) mph
   - Distance between the cities: 45 miles
   - Time until they meet: \(\frac{45 \text{ miles}}{30 \text{ mph}} = 1.5\) hours

2. **Distance Covered by Alicia:**
   - Alicia travels at 18 mph for 1.5 hours.
   - Distance: \(18 \text{ mph} \times 1.5 \text{ hours} = 27 \text{ miles}\)

3. **Verification Using Algebraic Method:**
   - Let \(x\) be the distance from City A where they meet.
     - Time taken by Alicia: \(\frac{x}{18}\)
   - Time taken by Beth: \(\frac{45 - x}{12}\)
   - Setting times equal: \(\frac{x}{18} = \frac{45 - x}{12}\)
   - Solving for \(x\):
     \[
     \frac{x}{18} = \frac{45 - x}{12} \implies 12x = 18(45 - x) \implies 12x = 810 - 18x \implies 30x = 810 \implies x = 27
     \]

Both methods confirm that the distance from City A when they meet is 27 miles.

\[
\boxed{27}
\]

Figure 22: Part 2 of the case of local score.

## Global Score Case

Please reason step by step, and put your final answer within \boxed{}.<|User|>Cities $A$ and $B$ are $45$ miles apart. Alicia lives in $A$ and Beth lives in $B$. Alicia bikes towards $B$ at 18 miles per hour. Leaving at the same time, Beth bikes toward $A$ at 12 miles per hour. How many miles from City $A$ will they be when they meet?<|Assistant|><think>

Okay, so I have this problem where City A and City B are 45 miles apart. Alicia lives in A and Beth lives in B. They both start biking towards each other at the same time. Alicia is biking at 18 miles per hour, and Beth is biking at 12 miles per hour. I need to figure out how many miles from City A they will be when they meet.

Hmm, let me think. So, they're starting from two points that are 45 miles apart. They're moving towards each other. So, their speeds are adding up because they're approaching each other. That makes sense. So, the combined speed at which they're closing the distance between them is Alicia's speed plus Beth's speed.

Let me write that down. Alicia's speed is 18 mph, Beth's speed is 12 mph. So, combined speed is 18 + 12, which is 30 mph. So, together, they're moving towards each other at 30 miles per hour.

Now, the total distance between them is 45 miles. So, I need to find out how long it takes for them to meet. Since they're moving towards each other at 30 mph, the time it takes to cover 45 miles together would be the time until they meet.

Time is equal to distance divided by speed, right? So, time = 45 miles / 30 mph. Let me calculate that. 45 divided by 30 is 1.5 hours. So, it will take them 1.5 hours to meet.

Okay, so now I know that they meet after 1.5 hours. The question is asking how many miles from City A they will be when they meet. So, I need to find out how far Alicia has traveled in that time because she started from City A.

Alicia's speed is 18 mph, so in 1.5 hours, she would have traveled 18 * 1.5 miles. Let me compute that. 18 times 1 is 18, and 18 times 0.5 is 9, so 18 + 9 is 27. So, Alicia has traveled 27 miles from City A when they meet.

Wait, let me double-check that. So, if Alicia has gone 27 miles and Beth has gone, let me calculate Beth's distance. Beth's speed is 12 mph, so in 1.5 hours, she goes 12 * 1.5. 12 times 1 is 12, and 12 times 0.5 is 6, so 12 + 6 is 18. So, Beth has gone 18 miles from City B.

Now, if we add Alicia's 27 miles and Beth's 18 miles, that's 27 + 18 = 45 miles. Which is exactly the distance between the two cities. That makes sense because they started 45 miles apart and together they covered that distance when they met. So, that checks out.

Another way to think about it: since they started at the same time and met at the same time, the time taken for both is the same. So, the distance each has covered plus the other's distance should equal 45 miles. So, Alicia's distance plus Beth's distance equals 45. Let me represent that as an equation.

Let t be the time in hours until they meet. Then:

Distance_Alicia = 18t

Distance_Beth = 12t

So, 18t + 12t = 45

Which simplifies to 30t = 45

So, t = 45 / 30 = 1.5 hours

Therefore, the distance from City A is 18 * 1.5 = 27 miles.

Yes, that seems consistent.

Wait, just to make sure I didn't make a mistake in my initial thinking. So, they're moving towards each other, so their relative speed is additive. That's correct because they are moving in opposite directions towards each other.

So, the combined speed is 18 + 12 = 30 mph. So, 45 miles divided by 30 mph is 1.5 hours. So, that's 1 hour and 30 minutes. So, in that time, Alicia would have gone 18 mph * 1.5 hours, which is 27 miles. And Beth would have gone 12 * 1.5 = 18 miles. So, 27 + 18 = 45, which is correct.

So, I think that's solid. So, the answer is 27 miles from City A.

Wait, let me think again if there's another way to approach this. Maybe using ratios or something else.

So, the speeds are 18 and 12, which can be simplified. Let's see, 18 and 12 have a common factor of 6, so dividing both by 6, we get 3 and 2. So, their speed ratio is 3:2.

Figure 23: Part 1 of the case of global score.

That means that for every unit of time, Alicia covers 3 parts and Beth covers 2 parts. So, together, they cover 5 parts. Since the total distance is 45 miles, each part is 45 / 5 = 9 miles. Therefore, Alicia covers 3 parts, which is 27 miles, and Beth covers 2 parts, which is 18 miles.

So, same answer. So, that also confirms that the meeting point is 27 miles from City A.

Another way to think about it is, since Alicia is faster, she will cover more distance. So, the ratio of their distances should be the same as the ratio of their speeds.

So, Alicia's speed is 18, Beth's speed is 12. So, the ratio is 18:12, which simplifies to 3:2. So, Alicia will have traveled 3/5 of the total distance, and Beth will have traveled 2/5.

Therefore, distance from City A is (3/5)*45 = 27 miles.

Yes, that's another way to get the same answer.

Alternatively, if I set up equations for their positions as functions of time and solve for when their positions are equal.

Let me try that approach too, just to make sure.

Let's denote t as the time in hours until they meet.

Alicia starts at City A, so her position at time t is 18t miles from A.

Beth starts at City B, which is 45 miles away from A, so her position at time t is 45 - 12t miles from A.

When they meet, their positions are equal. So, 18t = 45 - 12t

Solving for t:

18t + 12t = 45

30t = 45

t = 45 / 30

t = 1.5 hours, which is 1.5 hours.

So, Alicia has traveled 18 * 1.5 = 27 miles from City A.

Yes, that's consistent with what I found earlier.

So, whether I use the relative speed approach, the ratio of speeds, or setting up equations for their positions, I arrive at the same answer: 27 miles from City A.

I think that's solid. I don't see any mistakes in my reasoning. So, the answer should be 27 miles.

**Final Answer**
They will meet \boxed{27} miles from City $A$.
</think>

Cities \( A \) and \( B \) are 45 miles apart. Alicia bikes towards \( B \) at 18 miles per hour, and Beth bikes towards \( A \) at 12 miles per hour. We need to determine how many miles from City \( A \) they will be when they meet.

First, we calculate their combined speed:
\[
18 \text{ mph} + 12 \text{ mph} = 30 \text{ mph}
\]

Next, we find the time it takes for them to meet by dividing the distance between the cities by their combined speed:
\[
\text{Time} = \frac{45 \text{ miles}}{30 \text{ mph}} = 1.5 \text{ hours}
\]

In 1.5 hours, Alicia, traveling at 18 mph, will cover:
\[
18 \text{ mph} \times 1.5 \text{ hours} = 27 \text{ miles}
\]

Thus, they will meet 27 miles from City \( A \).
\[
\boxed{27}
\]

Figure 24: Part 2 of the case of global score.

