# OpenReview forum: "G-KV:  Decoding-Time  KV Cache Eviction  with Global Attention"
_ICLR.cc/2026/Conference — ICLR 2026 Conference Withdrawn Submission_

### Official Review · Reviewer_tGhE · 2025-10-17

**Soundness:** 2
**Presentation:** 3
**Contribution:** 3
**Rating:** 4
**Confidence:** 4

**Summary:**

This paper introduces G-KV, a decoding-time KV Cache eviction method designed to address the computational and memory bottlenecks that large language models face during long-sequence reasoning tasks. The authors argue that existing methods, which often rely on local attention scores to determine which tokens to evict, fail to capture the long-term importance of tokens. To solve this, G-KV introduces a "global score" that combines the current local attention score with historical scores, allowing for a more accurate assessment of a token's long-term value. Furthermore, the paper explores post-training techniques, including reinforcement learning and distillation, to better adapt the model to a compressed KV Cache environment. Experiments on the AMC and AIME mathematical reasoning benchmarks demonstrate that G-KV achieves significant performance improvements over existing methods.

**Strengths:**

1. The core motivation—that local attention is insufficient for capturing long-term token importance—is both intuitive and critical. The experiment in Section 4 (Figure 1), which convincingly demonstrates that the set of attended tokens shifts across different time windows, provides strong empirical support for the proposed “global score.”

2. I appreciate the paper’s writing style, especially the clarity of its figures and the *Observations* presented throughout, which make the motivation and overall argument easy to follow.

**Weaknesses:**

1. Limited Benchmark Coverage: The evaluation is primarily focused on mathematical reasoning tasks. While this effectively demonstrates the model's capabilities in long chain-of-thought scenarios, it may not fully prove the generalizability of the G-KV method to other types of long-text tasks. For instance, many state-of-the-art methods like SnapKV are also tested on benchmarks such as **LongBench** (for comprehensive long-text understanding) and **Needle-in-a-Haystack** (for long-range information retrieval). Including experiments on these more general long-context benchmarks would strengthen the paper's conclusions.

2. Insufficient Analysis of Method Overhead: The G-KV method introduces a "global score," which requires storing historical scores. Although the paper mentions in Appendix G that this overhead is negligible, it does not provide detailed empirical data to support this claim. Does calculating the global score introduce additional computational latency at each compression step? It would be beneficial for the authors to provide more specific experimental data, such as: **(1) What is the exact increase in memory (VRAM) usage for G-KV compared to methods that only use a local score? (2) On the same hardware and with the same batch size, what is the time cost of the G-KV algorithm itself (i.e., the scoring and sorting process)?** This would help readers more fully assess the method's efficiency.

**Questions:**

1. Regarding the benchmarks: Have you considered evaluating G-KV on more general long-context benchmarks like LongBench or Needle-in-a-Haystack? This would help validate your method's effectiveness on tasks beyond mathematical reasoning.

2. Regarding efficiency overhead: Could you provide more detailed data on the computational and memory overhead introduced by the global score calculation? Specifically, what is the added latency and extra memory consumption per compression step compared to a local-score-only method?

3. Regarding the hyperparameter $\alpha$: Figure 4 shows that the method's performance is quite sensitive to the decay factor $\alpha$. For new models or tasks, do you have any recommendations or heuristics for setting this hyperparameter, other than extensive experimental search?

4. Regarding token distribution: The finding in Figure 5 is very interesting—it shows G-KV retains tokens more uniformly across the entire sequence, whereas other methods are biased toward the end. Could you elaborate on why this uniform distribution is beneficial? Does it suggest that G-KV is better at preserving key information from the initial prompt, thus preventing context loss during long-range generation?

---

> ### Author Response · Authors · 2025-11-21
>
> Response to "Limited Benchmark Coverage"
>
> Thank you for the suggestion. We would like to clarify that these benchmarks are typically used to evaluate pre-filling stage token eviction methods, where the primary challenge lies in retrieving relevant content from long input sequences. In contrast, our method is specifically designed for inference-time scenarios, where the model must retain and manage information across generated tokens during decoding.
>
> Our focus aligns more closely with inference-oriented tasks, which is also the case for R-KV (NeurIPS 2023)—a related work that similarly evaluates only on inference-time reasoning benchmarks.
>
> That said, we have attempted to evaluate our method in the code generation domain (LiveCodeBench), and the experimental results are as follows:
>
>
>
>
> |              | 256        | 512        | 1024       | 1536       | 2048       |
> | ------------ | ---------- | ---------- | ---------- | ---------- | ---------- |
> | Full-KV      | 0.5806     |            |            |            |            |
> | StreamingLLM | 0.0200     | 0.0812     | 0.1950     | 0.2825     | 0.3250     |
> | Local Score  | 0.0800     | 0.1400     | 0.2412     | 0.3200     | 0.3700     |
> | SnapKV       | 0.0781     | 0.1406     | 0.2515     | 0.3281     | 0.3750     |
> | R-KV         | 0.1125     | 0.2212     | 0.3525     | 0.4312     | 0.4837     |
> | G-KV         | **0.1650** | **0.2537** | **0.3787** | **0.4387** | **0.4937** |
>
>
>
> From the experimental results, we observe that our method consistently outperforms the baselines across different budget settings.

---

> ### Author Response · Authors · 2025-11-21
>
> Response to "Insufficient Analysis of Method Overhead"
>
> Thank you for the insightful question. We appreciate the opportunity to clarify the overhead introduced by the global scoring mechanism in G-KV and provide additional empirical data to support our claims.
>
> We measured the average compression time for both the local score and global score strategies. Using the global score incurs approximately 5 ms of additional latency. However, the overall impact on performance is negligible, as compression accounts for only about 1% of the total decoding time.
>
>
> | budget                                | 512     | 1024    | 2048     |
> | ------------------------------------- | ------- | ------- | -------- |
> | Avg compression time (local score)    | 46.1 ms | 74.5 ms | 117.6 ms |
> | Avg compression time (global score)   | 50.5 ms | 79.4 ms | 121.4 ms |
> | compression time ratio (local score)  | 0.77%   | 1.10%   | 1.59%    |
> | compression time ratio (global score) | 0.83%   | 1.20%   | 1.57%    |
>
> The additional VRAM usage introduced by the global score grows proportionally with the size of the retained KV cache. For instance, in the case of Qwen-7B with a 2k token budget and a batch size of 128, the total KV cache occupies approximately 7 GB, while the overhead introduced by the global score is around 28 MB.

---

> ### Author Response · Authors · 2025-11-21
>
> We thank the reviewer for pointing out the issue of hyperparameter sensitivity. To better address this concern, we have conducted additional experiments focusing on the global decay coefficient **α**, which controls the influence of historical scores.
>
>
>
> As can be seen from the table, the performance is relatively stable between **0.8 and 0.9**, with fluctuations mostly around **2pp~3pp**, and this stability offers a **15pp~20pp** improvement compared to the local score. Its fluctuation is relatively small compared to the fluctuations caused by hyperparameters $\alpha$, so we recommend setting the $\alpha$between 0.8 and 0.9, as this range provides stable and good performance.
>
> |       |        |        |        |        |        |        |            |            |            |        |        |
> | ----- | ------ | ------ | ------ | ------ | ------ | ------ | ---------- | ---------- | ---------- | ------ | ------ |
> | alpha | 0      | 0.2    | 0.4    | 0.6    | 0.7    | 0.75   | 0.8        | 0.85       | 0.9        | 0.95   | 1.0    |
> | qwen  | 0.2804 | 0.3390 | 0.3546 | 0.3781 | 0.4000 | 0.4218 | **0.4375** | **0.4750** | **0.4710** | 0.5156 | 0.4601 |
> | llama | 0.3640 | 0.4156 | 0.4437 | 0.4937 | 0.4906 | 0.4968 | **0.5218** | **0.5312** | **0.5406** | 0.5187 | 0.4593 |

---

> ### Author Response · Authors · 2025-11-21
>
> Response to Question 4
>
> It is important to note that many of these early tokens (kept by global score) are selected by the attention mechanism itself, indicating that this portion of the context continues to be frequently referenced during later stages of reasoning. However, due to the intermittent nature of attention, local scoring strategies may fail to capture the long-term importance of such tokens, potentially leading to their early eviction. This, may negatively impact downstream reasoning or generation quality. In contrast, global score retains tokens from diverse positions in the sequence helps preserve a more complete and representative context, which is particularly beneficial for tasks involving multi-hop reasoning or long-range dependencies.

---

> ### Comment · Area_Chair_vBbM · 2025-11-23
>
> Dear reviewer,
>
> Thanks for your time and effort in reviewing ICLR2026 submissions. The authors have submitted their responses to your review. Please take the time to read and raise your further comments, and discuss with the authors.
>
> Best regards,
>
> AC

---

> > ### Comment · Reviewer_tGhE · 2025-11-23
> >
> > The authors' rebuttal has partially addressed my concerns. However, the authors have not provided a revised PDF manuscript that comprehensively incorporates the improvements suggested by reviewers. For instance, the missing baseline results mentioned in the reviews have not been adequately presented in an updated version, which would have better demonstrated the advantages of their method. Given that this is not my primary area of expertise, I intend to maintain my current score and await further discussion with other reviewers before making a final decision on whether to adjust my rating.

---

> > > ### Author Response · Authors · 2025-11-26
> > >
> > > Thank you for your comments. We have submitted a revised version of the manuscript, which incorporates the improvements discussed in our rebuttal. If there are any remaining concerns or suggestions, we would be more than happy to further discuss them with you. Thank you again for your time and feedback.

---

### Official Review · Reviewer_Uc4V · 2025-10-29

**Soundness:** 3
**Presentation:** 3
**Contribution:** 2
**Rating:** 4
**Confidence:** 5

**Summary:**

The authors propose a KV cache selection strategy that comprehensively considering both local and historical attention contexts to enhances accuracy after cache compression. Furthermore, they introduce an additional post-training optimization step designed to adapt the model to the compressed KV Cache, which subsequently yields further performance improvements.

**Strengths:**

- The paper is well-structured and easy to follow.
- The improvement achieved through the post-training optimization is noteworthy.

**Weaknesses:**

- The experimental results are unstable and require validation on a broader range of models, particularly larger-scale models (e.g., 32B and 70B parameters).
    * Specifically, while the performance gain for the DeepSeek-R1-Distill-Qwen-7B model in Table 1 appears acceptable, the improvements for the DeepSeek-R1-Distill-Llama-8B model in Table 2 are very inconsistent.
- The experimental results indicate that significant performance improvements are only achieved under very low cache budgets(like 256), which are likely impractical or unusable in real-world scenarios. The improvement is marginal in large budget scenarios.
- Lack of Novelty.
- The post-training optimization appears largely disconnected from the proposed global score mechanism, suggesting an ad hoc addition.
    - Fine-tuning a model to adapt to the sparsity inherent in cache compression is a general optimization technique and is not specifically tied to the authors' KV cache compression algorithm.
- The proposed method will likely impact the Time-to-First-Token(TTFT) performance. The authors need to include an experimental analysis of the TTFT overhead.

**Questions:**

- Performance and concern: In Table 4, the results show that G-KV (R-KV w/ global score ) achieves some performance gain over plain R-KV. Since the consideration of a "global score" is expected to introduce additional computational overhead, could the authors explain why G-KV still maintains an advantage?
- Data Inconsistency between Tables: The data presented in Table 1 and Table 3 does not align. Specifically, the "Untrained" baseline data in Table 3 does not match the "G-KV" data in Table 1. An explanation for this discrepancy is required.

---

> ### Author Response · Authors · 2025-11-21
>
> Response to Weaknesses1
>
> Thank you for the feedback. We would like to note that under constrained memory budgets, our method demonstrates a clear advantage on both Qwen-7B and LLaMA-8B. While the results on LLaMA-8B exhibit slightly variance, the performance remains competitive. We agree that evaluating on a broader range of models, especially larger-scale ones, would help further validate the generality of our method. However, due to the significant inference time required (often exceeding ten hours per model per setting) it is unfortunately infeasible for us to complete such large-scale evaluations within the discussion period. Nevertheless, we are actively working to include results on additional models, which we believe will provide further evidence of the robustness and scalability of our approach.

---

> ### Author Response · Authors · 2025-11-21
>
> Response to Weaknesses2
>
> These low cache budget settings are intentionally designed to stress-test the effectiveness of compression and eviction strategies. When memory is abundant, the differences between methods naturally diminish, as most tokens can be retained regardless of the policy. Therefore, low-budget scenarios offer a more direct and intuitive reflection of an algorithm’s effectiveness. Even under high cache budgets, an improvement of 1–2 percentage points is still considered a meaningful gain, especially given the already strong performance of baseline methods (close to Full KV).

---

> ### Author Response · Authors · 2025-11-21
>
> Response to "Lack of Novelty"
>
> While we understand the concern, we believe our work offers meaningful contributions that may not be immediately apparent. Our method is designed to be simple yet effective, and one of its key strengths lies in its broad compatibility with existing approaches. Rather than proposing entirely new architectures, our contribution focuses on delivering a practical and general solution that can be readily integrated into a wide range of existing systems. We believe this kind of universality and efficiency, especially under strict memory constraints, represents a valuable contribution to the field.

---

> ### Author Response · Authors · 2025-11-21
>
> Response to Weaknesses4
>
> It is true that the post-training technique is a general-purpose method and is not specifically designed for our algorithm. However, it is applied to enhance the performance of KV cache compression strategies, and it integrates closely and effectively with our approach. We believe that this technique remains valuable in the context of our work. In the revised version, we will make a clearer distinction between the two components and better articulate their relationship in the writing.

---

> ### Author Response · Authors · 2025-11-21
>
> Response to "include an experimental analysis of the TTFT"
>
> We would like to clarify that no additional latency is introduced when the prefilling length is within the memory budget, as our method does not perform any compression in that case. However, when the prefilling length exceeds the budget, compression is triggered before decoding begins, which may affect TTFT.
>
> To quantify this, we report the average compression time under different sequence lengths (batch size = 32):
>
> |   |   |   |   |
> |---|---|---|---|
> |length|512|1024|2048|
> |compression time|50.5 ms|79.4 ms|121.4 ms|
>
> As shown, the overhead scales moderately with sequence length and remains within a practical range for most real-world applications.

---

> ### Author Response · Authors · 2025-11-21
>
> Response to Question 1
>
> Thank you for the thoughtful question. First, in actual inference scenarios, padding tokens may be introduced, but we exclude padding tokens when measuring throughput. This may partially account for the differences observed in Table 4. Second, G-KV cancels the pooling operation which is used in SnapKV and R-KV. This simplification reduces computational overhead and may effectively offset the cost introduced by computing the global score.

---

> ### Author Response · Authors · 2025-11-21
>
> Response to Question 2
>
> This is because our training was conducted using a 4k context window, due to computational resource constraints. To ensure consistency, we also performed evaluation under the same 4k context length. This design explains the observed discrepancy. These details have been stated in the paper.

---

> ### Comment · Area_Chair_vBbM · 2025-11-23
>
> Dear reviewer,
>
> Thanks for your time and effort in reviewing ICLR2026 submissions. The authors have submitted their responses to your review. Please take the time to read and raise your further comments, and discuss with the authors.
>
> Best regards,
>
> AC

---

> ### Author Response · Authors · 2025-11-27
>
> Due to the high evaluation cost of inference tasks, it is challenging to conduct a complete evaluation on a 32B model. However, we have added evaluations of Qwen3-8B on AMC 23. Even under a 2k budget, our method achieves a 6pp improvement over R-KV, coming very close to the performance of Full-KV.
>
> pass@1
>
> |             | 256        | 512        | 1024       | 1536       | 2048       |
> | ----------- | ---------- | ---------- | ---------- | ---------- | ---------- |
> | Full-KV     | 0.9062     |            |            |            |            |
> | local score | 0.0500     | 0.2000     | 0.4437     | 0.5875     | 0.7062     |
> | SnapKV      | 0.0562     | 0.2625     | 0.4562     | 0.5937     | 0.7625     |
> | R-KV        | 0.1125     | 0.4500     | 0.7125     | 0.8000     | 0.8187     |
> | G-KV        | **0.3562** | **0.5562** | **0.7656** | **0.8406** | **0.8781** |
>
> Token Retention Rate
>
> |             | 256    | 512    | 1024   | 1536   | 2048   |
> | ----------- | ------ | ------ | ------ | ------ | ------ |
> | local score | 0.0164 | 0.0753 | 0.2272 | 0.3443 | 0.4286 |
> | SnapKV      | 0.0805 | 0.1003 | 0.2387 | 0.3566 | 0.4158 |
> | R-KV        | 0.0561 | 0.0959 | 0.1922 | 0.2936 | 0.3918 |
> | G-KV        | 0.0220 | 0.0645 | 0.1683 | 0.2744 | 0.3735 |
> We hope this addresses your concerns regarding the performance of G-KV across different models.

---

### Official Review · Reviewer_h1av · 2025-10-31

**Soundness:** 1
**Presentation:** 2
**Contribution:** 1
**Rating:** 2
**Confidence:** 4

**Summary:**

G-KV proposes a decoding-time KV-eviction rule that forms a per-token global score by combining a normalized local window score with an attenuated historical score (via a hard max), plus an optional RL fine-tuning objective; evaluation is largely math-reasoning with 7B/8B models.

**Strengths:**

The proposed global scoring mechanism is intuitive, which improves over R-KV at 256 tokens.

**Weaknesses:**

**Q1.**  The related-work section names several 2025 SOTA methods (R-KV, CAKE, KVzip) but they are not used as baselines in the main tables; RocketKV and ShadowKV from ICML’25, Raas ACL’25 are ignored for baseline comparison.

**Q2.** Novelty is incremental relative to temporal/union-aware and redundancy-aware methods.
 The global score (Eq. 3) is max(α·F_{t−1}, normalized S_t). a form of temporal accumulation via attenuation and a hard max. CAKE explicitly models temporal shifts and layer preferences; R-KV adds redundancy scores to avoid keeping near-duplicates; ShadowKV/KVzip reconstruct/repurpose KV context. G-KV’s “historical-local max” feels like a special case of these broader families and needs a stronger differentiation (e.g., theory or empirical wins against them).

**Q3.**  Authors state the RL objective “simplifies GRPO,” but the description aligns with REINFORCE w/ baseline; also, arguing the online setup “eliminates the need for clipping” is not supported—PPO/GRPO-style stability typically relies on clipping/trust-region constraints. Please justify with analysis or ablations. In general, I am not sure if, in practice, these RL work and deliver promising results.


**Q4.** Heuristic design issues:

The hard max in Eq. (3) may overreact to outliers; compare to (i) EMA / weighted average, (ii) max-pool w/ temperature, (iii) top-k smoothing. No ablation is shown.


Sensitivity to α, w (window), and s (interval) is missing; these hyper-params directly govern stability and recall. (Authors define them but don’t study sensitivity.)


**Q5.** Benchmarks are math-centric (AMC-23, AIME-24) with distilled 7B/8B models; The following ones are ignored GSM8K, MATH-500, CSQA, LiveCodeBench, and a long-context retrieval suite (LongBench/BabiLong).


**Q6.** Authors state decoding time is ~40% of Full-KV and ~90% memory reduction at 16k (appendices), but no system-level comparisons vs optimized kernels (e.g., FlashAttention-3 baselines in SeerAttention-R, or ShadowKV’s throughput).

**Q7.** Missing ablations/analyses

A) Hard-max design must be validated against smoother alternatives.
 Add an ablation that replaces the hard max with: (a) exponential moving average (EMA) of local and historical scores; (b) a convex combination with a learned or tuned mixing weight; and (c) a temperatured max/LogSumExp. For each, report pass@1, average KV-retention, and variance of retention across steps to assess robustness to attention spikes.

B) Aslo, the author should run structured sweeps: window size (w from {64, 128, 256}), compression interval (s from {8, 16, 32}), attenuation (α from {0.8, 0.9, 0.95, 0.99}), and budgets (b from {128, 256, 512, 1024, 2048}). Plot accuracy vs. retention and accuracy vs. decode-time to identify stable operating regions.

C) Head-wise analysis (are some heads persistently favored/evicted?) to contrast with HeadKV


**Typos:**

“PREILIMINARY” to  “PRELIMINARY”.

 (Eq. 2): The summation bounds (k=0 to w) imply w+1 elements, but the division is by w. This should be clarified (e.g., k=0 to w-1).

**Questions:**

I already indicated them in the weakness section.

---

> ### Author Response · Authors · 2025-11-21
>
> Response to Q1
>
>
> Thank you for highlighting this point. We selected R-KV (NeurIPS 2025) as the primary baseline in the main table because it is the most relevant and competitive SOTA method that supports permanent token eviction during the decoding phase. It also demonstrates strong performance across multiple reasoning tasks, making it an appropriate reference point for evaluating G-KV. In contrast, KVzip only compresses the input prompt, RocketKV fully preserves the KV cache during the decoding phase, and ShadowKV migrates the value state to the CPU. These methods are not consistent with the decoding phase eviction strategy.
>
>
>
> Regarding CAKE, we did not include a direct comparison since its use of dynamic token budgets is orthogonal to our method’s global scoring mechanism. However, we note that the motivation behind CAKE’s attention perturbation indicator bears some similarity to the intuition behind our global score. To further clarify their differences and potential connections, we conducted a comparison between the two techniques:
>
>
> |                    | Local Score | Attention Shift Score (CAKE) | Global Score (alpha=0.8) |
> | ------------------ | ----------- | ---------------------------- | ------------------------ |
> | qwen (budget 512)  | 0.2804      | 0.3125                       | **0.4375**               |
> | llama (budget 512) | 0.3640      | 0.4000                       | **0.5218**               |
>
>
>
> As shown in the table, our global score consistently outperforms both the local score and the attention shift score proposed in CAKE, across different base models. We will include these experimental results in the revised version of the paper.

---

> ### Author Response · Authors · 2025-11-21
>
> Response to Q2
>
> We acknowledge that the overall framework of our work builds upon prior efforts in the field. However, the core contribution of our method, the proposed global score, is a novel and independent technique. This reflects the strength of our approach. The global score is a general and flexible mechanism that is orthogonal to many existing techniques and can be easily integrated with them.
>
> Our experimental results show that combining the global score with prior methods such as SnapKV and R-KV leads to significant performance improvements, which demonstrates its effectiveness as a complementary component. In addition, the global score can also be extended to work with approaches such as HeadKV, AdaKV, and CAKE, which allocate token budgets dynamically across different attention heads or layers. We believe this compatibility makes the global score a valuable and adaptable component for broader integration in future research.

---

> ### Author Response · Authors · 2025-11-21
>
> Response to Q3
>
> Thank you for the comment. We believe the misunderstanding may stem from a difference in how we define the online setup. In our context, we define "online" as performing the policy update in a single batch after collecting experience, without reusing data across multiple updates or using mini batch (We allow micro-batching for gradient accumulation, where gradients are computed and accumulated across multiple micro-batches, but no model update is performed until the full batch has been processed). If the policy is updated over multiple mini batches, then the updated policy and the sampling policy become misaligned after the first batch. In that case, we consider it an offline setting.
>
> In our setting, the update is completed in a single batch, so the ratio $\frac{\pi_\theta}{\pi_{\theta_{\text{old}}}}$ is equal to 1. Under this condition, clipping mechanisms such as those used in PPO or GRPO become ineffective, as there is no divergence between the current and previous policies. This is the reasoning behind our claim that clipping is unnecessary in our specific setup.
>
> That said, we acknowledge that our original description may have caused confusion. To avoid ambiguity, we have revised the text in the paper to remove the statement that our objective "simplifies GRPO." Instead, we now present the standard GRPO objective directly. It is important to note that in the single-step update setting, our previously used objective is equivalent to GRPO. We appreciate the reviewer’s request for clarification and will make sure that these distinctions are clearly explained in the updated version.

---

> ### Author Response · Authors · 2025-11-21
>
> Response to Q4 and Q7 A) & B)
>
> Thank you for your suggestions. In fact, we initially experimented with a weighted average formulation, but found that it underperformed compared to the current max-based global score. Since our operation involves comparing only two values (the history and the current), techniques like temperature-based max pooling or top-k smoothing are less applicable in this setting.
>
>
>
> That said, motivated by your feedback, we further explored a sum-based formulation as an alternative to `max`, and conducted additional experiments comparing the following three variants of global score computation:
>
>
> - $\max( \alpha \times F_{t-1}[:,i], \frac{ S_t[:,i]}{\max_{j} S_{t}[:,j]} )$
> -  $\alpha \cdot F_{t-1}[:, i] + (1- \alpha) \frac{ S_t[:,i]}{\max_{j} S_{t}[:,j]} $
> - $\alpha \cdot F_{t-1}[:, i] + \frac{ S_t[:,i]}{\max_{j} S_{t}[:,j]} $
>
> The experimental results are as follows:
>
>
>
> | Alpha             | 0      | 0.2    | 0.4    | 0.6    | 0.7    | 0.75   | 0.8        | 0.85       | 0.9    | 0.95       | 1.0    |
> | ----------------- | ------ | ------ | ------ | ------ | ------ | ------ | ---------- | ---------- | ------ | ---------- | ------ |
> | Local Score       | 0.2804 |        |        |        |        |        |            |            |        |            |        |
> | Global Score（max） |        | 0.3390 | 0.3546 | 0.3781 | 0.4000 | 0.4218 | 0.4375     | 0.4750     | 0.4710 | **0.5156** | 0.4601 |
> | G-KV (mean)       |        | 0.3312 | 0.3250 | 0.3437 | 0.3781 | 0.4187 | 0.4218     | **0.4500** | 0.4281 | 0.4000     | 0.3437 |
> | G-KV (sum)        |        | 0.3312 | 0.3656 | 0.4031 | 0.4031 | 0.4125 | **0.4843** | 0.4812     | 0.4562 | 0.4593     | 0.4281 |
>
>
>
>
>
>
> |                     | 0      | 0.2    | 0.4    | 0.6    | 0.7    | 0.75   | 0.8        | 0.85   | 0.9        | 0.95   |        |
> | ------------------- | ------ | ------ | ------ | ------ | ------ | ------ | ---------- | ------ | ---------- | ------ | ------ |
> | Local Score         | 0.3640 |        |        |        |        |        |            |        |            |        |        |
> | Global Score（max）   |        | 0.3992 | 0.4250 | 0.4468 | 0.4812 | 0.5000 | 0.4632     | 0.4718 | **0.5218** | 0.5125 | 0.4820 |
> | Global Score (mean) |        | 0.3906 | 0.4593 | 0.4500 | 0.4531 | 0.4375 | **0.4718** | 0.4656 | 0.4593     | 0.4312 | 0.3718 |
> | Global Score(sum)   |        | 0.4156 | 0.4437 | 0.4937 | 0.4906 | 0.4968 | 0.5218     | 0.5312 | **0.5406** | 0.5187 | 0.4593 |
>
>
>
> The experimental results show that the mean-based formulation performs slightly worse than the max-based variant. While the max and sum variants achieve similar levels of performance, the sum formulation exhibits greater stability across runs. In our hyperparameter analysis, we observe that performance is relatively insensitive to the choice of α within the range of 0.8 to 0.9, and in most cases, values in this range yield the best results.
>
>
>
> The hyperparameters w (window) and s (interval) are widely used in prior work, and we adopt these established settings in our implementation to ensure consistency and fair comparison. While further tuning of these parameters may yield incremental performance gains, we believe such adjustments would not provide additional insight into the core contribution of our method, namely the global scoring mechanism. As a result, our sensitivity analysis primarily focuses on α, which directly influences the behavior and dynamics of the global score.

---

> ### Author Response · Authors · 2025-11-21
>
> Response to Q5
>
> Our method is specifically designed for long-form reasoning scenarios in inference-time models. We selected AMC-23 and AIME-24 as evaluation benchmarks because they represent competition-level mathematical tasks, where models must perform multi-step, extended reasoning to arrive at the correct answer.
>
>
>
> In contrast, math datasets such as GSM8K and MATH-500 are relatively more straightforward and may not sufficiently stress the reasoning capabilities we aim to evaluate. Similarly, tasks in LongBench and CSQA often involve shallower reasoning or retrieval-based skills, which are less aligned with our targeted use cases.
>
>
>
> That said, we agree that LiveCodeBench is a valuable and challenging benchmark, especially for evaluating reasoning in code generation tasks. To that end, we include our evaluation on LiveCodeBench using DeepSeek-R1-Distill (Qwen 7B) below:
>
>
>
>
> |              | 256        | 512        | 1024       | 1536       | 2048       |
> | ------------ | ---------- | ---------- | ---------- | ---------- | ---------- |
> | Full-KV      | 0.5806     |            |            |            |            |
> | StreamingLLM | 0.0200     | 0.0812     | 0.1950     | 0.2825     | 0.3250     |
> | Local Score  | 0.0800     | 0.1400     | 0.2412     | 0.3200     | 0.3700     |
> | SnapKV       | 0.0781     | 0.1406     | 0.2515     | 0.3281     | 0.3750     |
> | R-KV         | 0.1125     | 0.2212     | 0.3525     | 0.4312     | 0.4837     |
> | G-KV         | **0.1650** | **0.2537** | **0.3787** | **0.4387** | **0.4937** |
>
>
>
> From the experimental results, we observe that our method consistently outperforms the baselines across different budget settings.

---

> ### Author Response · Authors · 2025-11-21
>
> Response to Q6
>
> Thank you for raising this point. We would like to clarify that G-KV is a method for KV cache management, and is orthogonal to low-level attention kernel optimizations such as FlashAttention-3. In fact, we use the interface of  flash-attn in our implementation, if supported by the hardware, FlashAttention-3 can be enabled during inference. In addition, our study primarily aims to compare KV cache eviction strategies under a unified algorithmic framework. Our current implementation is built on the HuggingFace generation pipeline, which is commonly used in prior work but is not optimized for system-level efficiency. As a result, direct comparison with system-level optimized frameworks may not fully reflect the potential upper bound of our method. That said, our algorithm is fully compatible with inference engines such as vLLM, and we plan to explore integrating our method into such optimized inference systems as part of future work.

---

> ### Author Response · Authors · 2025-11-21
>
> Response to Q7 (c)
> Thanks for the suggestion regarding head-wise analysis. While we agree that such analyses may reveal interesting patterns, such as whether certain heads are consistently favored or evicted, they are not the primary focus of our work. That said, we recognize that head-specific scoring or budget allocation may offer further gains, and we consider this a promising direction for future work.

---

> ### Comment · Reviewer_h1av · 2025-11-24
>
> Thank the authors for providing some clarifications.
> I agree that KVzip is mainly focusing on prefill (though the authors tweak the SnapKV, which also primarily focuses on prefill for comparison). However, RocketKV and ShadowKV are relevant baselines. I am not sure what the authors mean by this statement: "RocketKV fully preserves the KV cache during the decoding phase". I believe RocketKV compresses KV cache by 400X (some in prefil and some in decode). Also, independent of utilizing the CPU, the authors should still be able to compare with ShadowKV. There are a few baselines specific to reasoning, like RaaS (ACL’25), HeadKV (ICLR’25), etc.  A claim of SOTA requires comparison against the leading methods in low and high token budgets, regardless of the underlying mechanism. Comparing with only one valid baseline (SnapKV and StreamingLLMs are weak baselines)  makes the comparison section weak. R-KV is also not very efficient in terms of throughput and even latency reduction.
> In addition, the fundamental concept of using historical attention information is used in many previous works. The main contribution is the specific formulation of the global score (Eq. 3). This is a refinement of how attention history is accumulated and normalized.
> I keep my score.

---

> > ### Author Response · Authors · 2025-11-24
> >
> > Regarding **RocketKV**, we agree that it reduces computational costs during decoding by applying **sparse attention** to selectively attend to a subset of the KV cache. However, it is important to emphasize that **RocketKV retains the full KV cache in memory during the decoding phase**, and thus **does not achieve memory savings** during this stage.
> >
> > As for **HeadKV**, we highlighted that our proposed method is **orthogonal** to theirs and can be **combined** with HeadKV. Therefore, we did not perform a direct comparison. Nonetheless, we observe that incorporating our **global score** into baselines such as **SnapKV** and **R-KV** consistently enhances their performance, demonstrating both the generality and effectiveness of our approach.
> >
> > > “R-KV is also not very efficient in terms of throughput and even latency reduction.”
> >
> > Could you kindly clarify the basis for this assertion? In fact, R-KV achieves significantly higher throughput compared to prior methods that only compress the prompt.
> >
> > Regarding the use of **historical attention information**, we would greatly appreciate it if you could point us to the **specific prior works** you are referencing. To the best of our knowledge, this formulation and its integration into real-time decoding pipelines have not been addressed in prior literature.

---

> > > ### Comment · Reviewer_h1av · 2025-11-24
> > >
> > > RocketKV has optimization in both prefill and decode. It is irrelevant whether the memory saving occurs during the prefill or decoding stage. As long as the memory is reduced  (e.g., 32% percent reported in RocketKV) and the method compresses the KV cache (e.g., 400X reported) while it reaches high accuracy, G-KV should demonstrate superiority over it to substantiate a SOTA claim. That is the same for other related works, such as HeadKV or Raas. Simply saying some methods are orthogonal will not justify the lack of comparison. Not sure if some token selection criteria are changed, the method can reach the same or higher accuracy as expected.  As I mentioned, comparing with only one relevant baseline (R-KV) makes the comparison section very weak.
> > >
> > > R-KV primary throughput improvement stems from enabling significantly larger batch sizes due to memory savings, rather than improving single-batch latency.  In latency-sensitive applications (small-batch inference), the overhead of dynamic eviction strategies (scoring, normalization, dynamic re-indexing) often dominates. Also, R-KV doesn't support flashattention and vllm, which makes the baseline not promising. The R-KV evaluation relies on pass@1 derived from  64 responses per question. This methodology is computationally expensive for evaluation. Furthermore, community feedback often notes discrepancies where reference implementations may default to greedy decoding (do_sample=False), despite what is stated in the paper. Please check GitHub issues on the R-KV paper.
> > >
> > > While G-KV's specific formulation (Eq. 3) differs from H2O's and other similar papers (additive accumulation), the core concept of using historical attention patterns to estimate long-term importance during dynamic eviction comes from these papers.

---

> ### Author Response · Authors · 2025-11-25
>
> Thank you for your follow-up discussion. We appreciate the opportunity to further clarify these points. RocketKV only uses SnapKV to compress the KV cache during the first stage. In the second stage, **all** KV cache entries generated during decoding are still retained; the reduction comes solely from sparse attention that lowers memory **access**, not memory **storage**. Thus, the claim of “400× reduction” specifically refers to memory traffic, while storage remains unchanged in stage two. RocketKV-MT corresponds exactly to this second stage. As stated explicitly in the RocketKV paper:
>
> > _“RocketKV-MT achieves the same memory traffic savings as RocketKV but does not introduce memory storage savings.”_
>
> Regarding **Head-KV**, it supports compression only during the prefilling stage and does not apply compression during decoding, which is why it was not included in our comparison. As for _Raas_, we were unable to locate it using the abbreviation. Could you kindly provide a citation so we can better understand and evaluate the method?
>
> Your comment that R-KV shows throughput advantages only under large batch sizes also appears to be based on a misunderstanding. With a fixed-length KV cache, R-KV’s attention complexity does not grow with the sequence length, leading to substantial speed-ups. Concerning the statement that “the overhead of dynamic eviction strategies often dominates,” we have not found evidence supporting this. Compression is triggered only at specific intervals, and our measurements show that the total compression cost is roughly **1%** of overall decoding time—introducing negligible overhead.  Furthermore, even under small batch sizes, our method (as the same as R-KV ) achieves an average decoding time of approximately 2/5 of that required for full-KV, which clearly demonstrates that R-KV retains considerable speed-up advantages in such scenarios.
>
> The claim that R-KV does not support FlashAttention is also inaccurate; as we previously clarified, both R-KV and our method fully support FlashAttention.
>
> On the concern about evaluation cost: **pass@1** is widely recognized as a standard metric for reasoning tasks, and we believe its use requires no special justification. If you see a specific issue with pass@1 or believe there is a more suitable metric for this setting, we would sincerely appreciate further clarification. Similarly, R-KV’s default greedy sampling strategy is simply a configuration choice and does not materially affect its validity as a baseline. Raising such points seems unnecessary and distracts from the core scientific discussion. We continue to regard R-KV as a **strong and appropriate** baseline, and the claim that it is “weak” appears unwarranted.
>
> Regarding **H2O**, this method tends to assign relatively higher scores to tokens in earlier positions due to their attention being accumulated more frequently. We explicitly address this limitation by introducing a decay factor α, and we have shown that H2O is essentially equivalent to a sum-based global score with α = 1, which indeed yields  worse performance in our experiments. This key distinction has been explicitly discussed in our paper.  Beyond H2O, we are not aware of other works adopting a similar approach.  If there are relevant references, we would appreciate it if you could provide them to support your claims.

---

> > ### Comment · Reviewer_h1av · 2025-11-28
> >
> > Thanks for the further explanation and clarification.
> >
> > I think the authors did not get my main point. My point is that, independent of where memory storage, memory access, and latency occur, authors need to show that their proposed method can outperform the alternatives. From the user’s perspective, it is not important whether you compress during prefill or decode; the user only cares about low memory footprint and access cost, with negligible accuracy loss.
> >
> > Other relevant papers are:
> >
> > RaaS: Reasoning-Aware Attention Sparsity for Efficient LLM Reasoning
> >
> > Dialogue Without Limits: Constant-Sized KV Caches for Extended Responses in LLMs (for your question about prefill vs. decode)
> >
> > Thanks for the clarification on R-KV.
> > I mentioned that R-KV can be considered a relevant baseline. However, authors still need to compare against a few more baselines to demonstrate the superiority of the proposed method. You can also check all reported issues here: https://github.com/Zefan-Cai/R-KV/issues
> >
> >
> > To my understanding, H2O introduced the core concept of using historical, accumulated attention patterns for dynamic eviction. G-KV mainly modifies the accumulation function (from sum to a decayed max). I do not consider this to be a major contribution.
> >
> > I change my score from reject to weak reject in case other reviewers have different opinions about the paper.

---

> > > ### Author Response · Authors · 2025-11-28
> > >
> > > Thank you for your response. The "Dialogue Without Limits: Constant-Sized KV Caches for Extended Responses in LLMs" paper introduces **MorphKV**, which is equivalent to the **local score** described in our paper. We will revise our paper to update the description of this baseline. Below is an efficiency comparison between our method and this approach. While our method introduces only a minimal delay (approximately 5 ms for each compression step), it achieves significantly improved performance.
> > >
> > > time for each compression step
> > >
> > > | budget  | 512     | 1024    | 2048     |
> > > | ------- | ------- | ------- | -------- |
> > > | MorphKV | 46.1 ms | 74.5 ms | 117.6 ms |
> > > | G-KV    | 50.5 ms | 79.4 ms | 121.4 ms |
> > >
> > > ratio  of compression time to total decoding time
> > >
> > > | budget  | 512   | 1024  | 2048  |
> > > | ------- | ----- | ----- | ----- |
> > > | MorphKV | 0.77% | 1.10% | 1.59% |
> > > | G-KV    | 0.83% | 1.20% | 1.57% |

---

> ### Author Response · Authors · 2025-11-27
>
> Methods such as **HeadKV**, which focus on prompt compression, **Quest**, which employs top-$k$ attention mechanisms, and **RocketKV**, which combines prompt compression with top-$k$ attention, differ fundamentally from our approach. In inference scenarios, the input may consist of only a few hundred tokens, while the output can extend to tens of thousands of tokens. In such cases, the KV cache of output becomes the primary source of GPU memory consumption. The aforementioned methods do not compress the KV cache of output, which limits their concurrency capacity. In contrast, our method maintains nearly constant memory usage throughout the decoding process, significantly enhancing concurrency, particularly in inference scenarios. Moreover, our evaluations demonstrate that with a $2k$ budget, G-KV achieves performance close to full KV. While we do not compare our method with these approaches, we will explicitly discuss the distinctions between methods such as **Quest** and **RocketKV** and our approach in the final version of this paper.

---

### Official Review · Reviewer_ziNN · 2025-11-04

**Soundness:** 3
**Presentation:** 3
**Contribution:** 2
**Rating:** 4
**Confidence:** 4

**Summary:**

This paper proposes G-KV, a decoding-time KV cache eviction method that introduces a global scoring mechanism combining historical and local attention scores to better preserve long-term token importance. It further enhances performance via post-training techniques—reinforcement learning with sparse attention masks and knowledge distillation. The method is evaluated on challenging mathematical reasoning benchmarks (AMC-23, AIME-24) using strong reasoning LLMs, showing significant gains over prior SOTA, especially under tight KV cache budgets (e.g., +19.15% pass@1 at 256 tokens).

The work is well-motivated, empirically solid, and addresses a critical bottleneck in long-context reasoning LLMs. The global scoring idea is simple yet effective, and the integration with existing methods (e.g., R-KV) demonstrates strong generalizability. The training-aware extensions (RL-Sparse, Distill) are thoughtfully designed to close the train-inference gap.
However, while the empirical results are compelling, the technical novelty is incremental, and several methodological and evaluation concerns limit the strength of the contribution for a top-tier conferences.

**Strengths:**

**Strong Empirical Results:**
  - Clear and consistent improvements over strong baselines across multiple models (Qwen-7B, Llama-8B), datasets (AMC, AIME), and KV budgets.
  - Gains are especially pronounced under low budgets (256–512 tokens), which are practically relevant for deployment.

**Insightful Motivation via Empirical Observation:**
  - Figure 1 provides compelling evidence that token importance is non-stationary across decoding windows—justifying the need for global scoring.
  - The analysis of token retention distribution (Figure 5) convincingly shows that G-KV preserves more diverse context than local methods.

**Practical and Modular Design:**
  - The global score is a drop-in replacement for local scores in existing eviction frameworks (H2O, SnapKV, R-KV).
  - Minimal computational overhead (confirmed by throughput results in Table 4).

**Comprehensive Evaluation:**
  - Includes efficiency metrics (throughput, memory, decoding time), ablation on α and λ, cross-model validation, and qualitative case studies (Appendix J).
  - Post-training methods (RL-Sparse, Distill) are well-motivated and show meaningful gains.

**Reproducibility:**
  - Code, distilled data, and environment configs are provided.

**Weaknesses:**

**Limited Technical Novelty:**
   - The global score (Eq. 3) is essentially an exponentially weighted moving average (EWMA) of normalized attention scores—a well-known technique in online learning and signal processing. While effective, it lacks deep algorithmic innovation.
  - The core idea resembles “heavy-hitter” tracking with decay, similar in spirit to H2O but with historical memory.

**Evaluation Scope is Narrow:**
  - Experiments are restricted to mathematical reasoning on two datasets. No evaluation on general QA, coding, or open-ended generation.
  - All models are distilled reasoning models from DeepSeek-R1. Performance on standard LLMs (e.g., Llama-3, Qwen2) or non-reasoning tasks is unverified.
  - No comparison to non-eviction compression methods (e.g., quantization, low-rank) that may offer better trade-offs.

**Hyperparameter Sensitivity:**
  - Performance heavily depends on α (decay) and λ (redundancy weight). While tuned, the paper doesn’t provide robustness analysis or automatic tuning strategies.
 - The optimal α ≈ 0.8–1.0 suggests historical scores dominate—raising questions about the marginal utility of local scores.

**Ambiguity in Training Protocol:**
  - RL-Sparse uses a sparse attention mask during training, but it’s unclear how this interacts with RoPE (rotary embeddings), which assumes full positional context. This could introduce positional bias.
  - Distillation uses teacher outputs from full KV, but student trains with sparse attention—a distributional mismatch not fully addressed.

**Claimed SOTA May Be Overstated:**
  - The 19.15% improvement is vs. R-KV under 256 tokens—but R-KV itself may not be the strongest baseline (e.g., CAKE, LightThinker are mentioned but not compared in main results).
  - No comparison to recent methods like StreamingLLM or Infini-Attention that handle long contexts differently.

**Questions:**

- How does G-KV perform on non-reasoning tasks (e.g., narrative continuation, summarization) or with standard LLMs (e.g., Llama-3-8B) without reasoning distillation?

- Besides math related task, what about those extremely long context reasoning task in coding domain like SWE bench? Besides, could you provide a computing and memory cost for different context length such as 4K, 8K, 16K, 32K, 64K, etc?

- Since G-KV evicts early tokens, how does the model handle positional information for retained early tokens under RoPE?

---

> ### Author Response · Authors · 2025-11-21
>
> Thank you for the insightful comments. We appreciate the reviewer’s interest in clarifying the source of novelty in our method.
>
> Response to "Limited Technical Novelty":
>
> Eq. (3) is not intended to introduce a new decay formulation; the contribution lies in how the global score is used for KV eviction. Prior approaches  implicitly assume that token importance can be inferred solely from the current observation window. However, our empirical analysis (Fig. 1) reveals that attention is highly intermittent, and tokens that appear unimportant locally often become crucial later.
> Accordingly, our score is not a standard EWMA: it is updated only for tokens that survive eviction, making the history conditioned on the evolving sparse KV state. This survival-dependent update allows the system to retain temporarily dormant but future-relevant tokens, which local or heavy-hitter strategies inevitably discard. Consequently, G-KV produces different retention behavior (Fig. 5) and achieves substantial gains under tight budgets.
>
> We believe such insights, simple yet changing what information is preserved during reasoning, hold meaningful value to the community.

---

> ### Author Response · Authors · 2025-11-21
>
> Response to "Evaluation Scope is Narrow":
>
> We explain that our method is designed to address caching and computational bottlenecks that arise during long-form decoding in reasoning tasks. We would like to kindly point out that this motivation is discussed in the introduction section of our paper. These challenges are particularly relevant to scenarios involving lengthy outputs inference, which G-KV aims to handle effectively.
> Therefore, we chose mathematical  tasks (AMC23 and AIME24) as our main evaluation benchmarks. These tasks, characterized by long output lengths and extreme sensitivity to contextual information preservation, are ideal scenarios for evaluating the effectiveness of caching compression methods during the decoding phase. We believe that the significant improvements achieved on these tasks (especially under low-budget settings) fully demonstrate the effectiveness and practical value of our method.
> We fully recognize the importance of evaluating G-KV on broader tasks. In response to this, we are actively adapting and testing G-KV on LiveCodeBench, a benchmark where code generation tasks inherently involve complex reasoning. The experimental results (DeepSeek-R1-Distill Qwen 7B) are as follows:
>
>
>
> |              | 256        | 512        | 1024       | 1536       | 2048       |
> | ------------ | ---------- | ---------- | ---------- | ---------- | ---------- |
> | Full-KV      | 0.5806     |            |            |            |            |
> | StreamingLLM | 0.0200     | 0.0812     | 0.1950     | 0.2825     | 0.3250     |
> | Local Score  | 0.0800     | 0.1400     | 0.2412     | 0.3200     | 0.3700     |
> | SnapKV       | 0.0781     | 0.1406     | 0.2515     | 0.3281     | 0.3750     |
> | R-KV         | 0.1125     | 0.2212     | 0.3525     | 0.4312     | 0.4837     |
> | G-KV         | **0.1650** | **0.2537** | **0.3787** | **0.4387** | **0.4937** |
>
>
> From the experimental results, we observe that our method consistently outperforms the baselines across different budget settings.
> As for non-elimination compression methods such as quantization and low-rank factorization (e.g., LoRA), their target dimensions differ from our method: G-KV focuses on dynamic retention decisions at the token granularity, while quantization and structural compression focus on the storage precision of the cached representation or the structural encoding method; the two are highly complementary.

---

> ### Author Response · Authors · 2025-11-21
>
> Response to "Hyperparameter Sensitivity"
>
>
>
> We thank the reviewer for pointing out the issue of hyperparameter sensitivity. To better address this concern, we have conducted additional experiments focusing on the global decay coefficient **α**, which controls the influence of historical scores.
>
>
>
> As can be seen from the table, the performance is relatively stable between **0.8 and 0.9**, with fluctuations mostly around **2pp~3pp**, and this stability offers a **15pp~20pp** improvement compared to the local score. Its fluctuation is relatively small compared to the fluctuations caused by hyperparameters $\alpha$, so we recommend setting the $\alpha$
> between 0.8 and 0.9, as this range provides stable and good performance.
>
> |       |        |        |        |        |        |        |            |            |            |        |        |
> | ----- | ------ | ------ | ------ | ------ | ------ | ------ | ---------- | ---------- | ---------- | ------ | ------ |
> | alpha | 0      | 0.2    | 0.4    | 0.6    | 0.7    | 0.75   | 0.8        | 0.85       | 0.9        | 0.95   | 1.0    |
> | qwen  | 0.2804 | 0.3390 | 0.3546 | 0.3781 | 0.4000 | 0.4218 | **0.4375** | **0.4750** | **0.4710** | 0.5156 | 0.4601 |
> | llama | 0.3640 | 0.4156 | 0.4437 | 0.4937 | 0.4906 | 0.4968 | **0.5218** | **0.5312** | **0.5406** | 0.5187 | 0.4593 |

---

> ### Author Response · Authors · 2025-11-21
>
> Response to "Ambiguity in Training Protocol"
>
> Regarding the interaction with RoPE, we would like to clarify that G-KV retains full positional context because RoPE is encoded into the KV cache using the original token positions in the sequence, not the compresed cache length. As such, even when sparse attention is applied, the positional embeddings remain aligned with the full sequence, thereby avoiding positional bias.
>
> Regarding the potential mismatch between the teacher and student distributions during distillation, we would like to clarify that although the teacher and student operate under different attention patterns (full vs. sparse KV), the key problem we aim to address is the mismatch between training and inference conditions. In our setup, the student is trained under the same constraints it will face at inference, thereby ensuring distributional consistency. The goal of distillation in this context is to enable the student to match the performance of a full-KV model, even under limited KV access, by effectively learning to approximate the teacher’s behavior within a compressed attention space.

---

> ### Author Response · Authors · 2025-11-21
>
> Response to the selection of baseline
>
> Thanks for raising this point. We would like to clarify that LightThinker requires large-scale training and is not suitable for comparison with training-free methods. As for StreamingLLM, we did include it in our extended experiments.
>
>
>
>
> |         |       |             | 256    | 512    | 1024   | 1536   | 2048   |
> | ------- | ----- | ----------- | ------ | ------ | ------ | ------ | ------ |
> | AMC 23  | Qwen  | SteamingLLM | 0.0250 | 0.1500 | 0.4000 | 0.5687 | 0.6562 |
> |         |       | Local Score | 0.1250 | 0.2804 | 0.5539 | 0.6835 | 0.7335 |
> |         | LLaMA | SteamingLLM | 0.0562 | 0.1875 | 0.3750 | 0.5437 | 0.5812 |
> |         |       | Local Score | 0.2031 | 0.3640 | 0.5960 | 0.7117 | 0.7632 |
> | AIME 24 | Qwen  | SteamingLLM | 0      | 0.0083 | 0.0666 | 0.1333 | 0.2166 |
> |         |       | Local Score | 0.0041 | 0.0156 | 0.1177 | 0.2114 | 0.2750 |
> |         | LLaMA | SteamingLLM | 0      | 0      | 0.0416 | 0.0666 | 0.1500 |
> |         |       | Local Score | 0.0104 | 0.0583 | 0.1364 | 0.2385 | 0.3479 |
>
>
>
> StreamingLLM performed poorly in our evaluation, yielding results that were even lower than those of the simplest local score baseline. We will include these experimental results in the revised version of the paper.
>
>
>
> Regarding CAKE, we did not include a direct comparison since its use of dynamic token budgets is orthogonal to our method’s global scoring mechanism. However, we note that the motivation behind CAKE’s attention perturbation indicator bears some similarity to the intuition behind our global score. To further clarify their differences and potential connections, we conducted a comparison between the two techniques:
>
>
>
>
> |                    | Local Score | Attention Shift Score (CAKE) | Global Score (alpha=0.8) |
> | ------------------ | ----------- | ---------------------------- | ------------------------ |
> | qwen (budget 512)  | 0.2804      | 0.3125                       | **0.4375**               |
> | llama (budget 512) | 0.3640      | 0.4000                       | **0.5218**               |
>
>
>
> As shown in the table, our global score consistently outperforms both the local score and the attention shift score proposed in CAKE, across different base models. We will include these experimental results in the revised version of the paper.

---

> ### Author Response · Authors · 2025-11-21
>
> Response to Question 1
>
> As described before, our approach is primarily designed for reasoning models and tasks, with a particular focus on addressing efficiency challenges during long-form decoding. We appreciate the opportunity to clarify this and will make sure to emphasize it more clearly in the final version of the paper.

---

> ### Author Response · Authors · 2025-11-21
>
> Response to Q2:
>
> For long-context tasks, the prefilling computation cost remains identical to that of full-KV attention, but our method introduces an additional compression step afterward.
>
>
> | budget                   | 512    | 1024   | 2048   |
> | ------------------------ | ------ | ------ | ------ |
> | avg compression time (s) | 0.0505 | 0.0794 | 0.1214 |
>
>
>
> As shown in the table above, the compression time varies slightly across different context lengths, with longer sequences requiring more time for compression. However, it is important to note that only the first compression step operates over the full long sequence. Subsequent compressions are applied over much shorter segments, typically of length budget + interval, which significantly reduces the recurring computational overhead.
>
>
>
> The memory cost of our method is determined by the given budget, as detailed in Appendix G. Regardless of the original context length, once compression is applied, the memory usage per request remains fixed under a fixed budget, ensuring predictable and bounded resource consumption during inference.

---

> ### Author Response · Authors · 2025-11-21
>
> Response to Q3
>
> Thank you for the question. We would like to clarify that we use the actual token position when applying RoPE (rotary position embeddings), regardless of the current KV cache length.
>
> For example, if the total sequence length reaches 8000 tokens, and the budgeted KV cache retains only 2048 tokens, the position index of the next token is still 8001. Since RoPE is encoded into the KV cache using the true positional indices, the eviction of earlier tokens does not affect the correctness of positional encoding for the retained tokens. This ensures that the model continues to receive accurate positional signals throughout the sequence, even under compression.

---

> ### Comment · Area_Chair_vBbM · 2025-11-23
>
> Dear reviewer,
>
> Thanks for your time and effort in reviewing ICLR2026 submissions. The authors have submitted their responses to your review. Please take the time to read and raise your further comments, and discuss with the authors.
>
> Best regards,
>
> AC

---

> > ### Comment · Reviewer_ziNN · 2025-11-28
> >
> > I appreciate the authors’ detailed responses, which address some of my concerns; however, I still believe the insight offered to the community is relatively marginal, so I am maintaining my original score.

---

### Author Response · Authors · 2025-11-24
**Manuscript revision**

We sincerely thank the reviewers  for your meticulous review and valuable suggestions. These insightful comments have been instrumental in improving the quality of our paper and enhancing the clarity of its presentation.  After carefully reading and fully understanding the review comments, we have conducted a thorough revision of the paper. The main updates are as follows:

1. Two additional forms of global score have been introduced, and a systematic experimental analysis has been included in Section 7.2.
2. Equation (6) has been revised to avoid potential ambiguities or misunderstandings.
3. Experimental results on the coding task from LiveCodeBench have been added (see Section 7.3) to further validate the generality of our method.
4. StreamingLLM has been introduced as a new baseline to enhance the comprehensiveness and fairness of the comparisons.
5. A statistical analysis of the delay overhead introduced by the global score has been added in Section 7.5 to supplement the evaluation from an efficiency perspective.
6. Comparative experiments with techniques similar to those in CAKE have been conducted (see Section 7.2) to better highlight the uniqueness and advantages of our method.
7. The structure of the paper has been optimized and reorganized: the content in Section 7.2 regarding integration with other methods has been moved to Appendix D, and experiments related to LLaMA have been relocated to Appendix H to improve the focus and logical flow of the main text.
8. Replace the tabular presentation of experimental results with graphical representations for more intuitive comparisons.

We deeply appreciate the efforts and support of the reviewers and AC, and we hope that this revision better addresses the concerns and suggestions raised by the reviewers.

---

### Note · Authors · 2026-01-05

**Comment:**

Dear Reviewers and Area Chair,

We sincerely thank you for your time and constructive feedback on our paper. We greatly appreciate the thoughtful reviews and valuable suggestions provided during the evaluation process.

After careful consideration, we have decided to withdraw our submission in order to make substantial improvements to the paper. We believe this will allow us to further enhance the quality of our work.

**Withdrawal Confirmation:**

I have read and agree with the venue's withdrawal policy on behalf of myself and my co-authors.